# PEAK1 maintains tight junctions in intestinal epithelial cells and resists colitis by inhibiting autophagy-mediated ZO-1 degradation

Zaikuan Zhang[1,2,10], Yajun Xie [1,3,10] ✉, Qiying Yi[4,10], Jianing Liu [5,6,10], Lin Yang[1], Runzhi Wang[7], Jing Cai[7], Xinyi Li[1], Xiaosong Feng[1], Shixiang Yao[8], Zheng Pan[9], Magdalena Paolino [5,6] ✉ & Qin Zhou [1,7] ✉

Tight junctions are crucial for maintaining intestinal barrier homeostasis, but how organisms modulate these junctions remain unclear. Here, we show a role for PEAK1 at cell-cell contact sites, where it interacts with ZO-1 via a conserved region spanning amino acids 714-731. This interaction masks the LC3-interacting region on ZO-1, preventing autophagy-mediated ZO-1 degradation and preserving the integrity of tight junctions in intestinal epithelial cells. Src-mediated phosphorylation of PEAK1 at Y724 promotes the binding between PEAK1 and ZO-1 to stabilize ZO-1 in intestinal epithelial cells. Additionally, PEAK1 binds to CSK to positively regulate Src activity. Loss of PEAK1 in intestinal epithelial cells leads to decreased Src activity and lower ZO-1 protein levels, resulting in disrupted tight junctions, both in vitro and in vivo. In mice, *Peak1* deficiency increases intestinal epithelium permeability and exacerbates inflammation in experimentally induced colitis models. Our findings reveal PEAK1's critical role in maintaining tight junction integrity and resistance to intestinal inflammation, extending its known function from promoting tumor cell proliferation and migration to essential physiological processes. These insights refine our understanding of the mechanisms regulating tight junctions and offer potential therapeutic targets for enhancing epithelial barrier function and treating related diseases.

The intestinal epithelium acts as a barrier between the external environment and the tightly regulated internal milieu and is essential for human health. Increased gut permeability is associated with various intestinal diseases, including inflammatory bowel disease (IBD), celiac disease, and irritable bowel syndrome (IBS)[1]. Human studies have shown that loss of the intestinal barrier function and increased intestinal permeability are also associated with a higher risk of developing Crohn's disease[2–4]. Thus, defective intestinal barrier function is a biomarker of IBD. While recent research has identified some causes of defective intestinal barrier function[5–7], the underlying molecular mechanisms remain to be elucidated.

[1]The Ministry of Education Key Laboratory of Laboratory Medical Diagnostics, the College of Laboratory Medicine, Chongqing Medical University, Chongqing, P. R. China. [2]Chongqing University Three Gorges Hospital, Chongqing University, Chongqing, P. R. China. [3]Western Institute of Digital-Intelligent Medicine, Chongqing, P. R. China. [4]The Experimental Animal Center, Chongqing Medical University, Chongqing, P. R. China. [5]Department of Medicine Solna, Center for Molecular Medicine, Karolinska Institutet, Stockholm, Sweden. [6]Karolinska University Hospital, Stockholm, Sweden. [7]The School of Basic Medical Sciences, Harbin Medical University, Harbin, P. R. China. [8]The College of Food Science, Southwest University, Chongqing, P. R. China. [9]The College of Traditional Chinese Medicine, Chongqing Medical University, Chongqing, P. R. China. [10]These authors contributed equally: Zaikuan Zhang, Yajun Xie, Qiying Yi, Jianing Liu. ✉e-mail: yjxie@cqmu.edu.cn; magdalena.paolino@ki.se; zhouqin@hrbmu.edu.cn

Tight junctions between intestinal epithelial cells play a central role in the intestinal barrier. The organization of tight junctions relies on protein-protein interactions via conserved domain binding and post-translational modifications. The zonula occluden family proteins ZO-1, ZO-2, and ZO-3, use multiple domains, including PSD-95/Dlg/ZO-1 (PDZ), Src homology-3 (SH3), actin binding region (ABR) and ZU5 domains, to recruit and assemble tight junction proteins[8–11]. The specific knockout of ZO-1 in intestinal epithelia cells disrupts microvillous and increases macromolecular permeability[12]. In cultured epithelial cells, deficiencies in ZO-1 and ZO-2 prevent claudin recruitment to tight junctions, impairing barrier function[10]; these defects can be rescued by expression of full-length ZO-1 or a PDZ domain-containing ZO-1 truncation[13], underscoring ZO-1's role in maintaining tight junction integrity. However, the mechanisms controlling ZO-1 protein stability are largely unknown.

Post-translational modifications, particularly phosphorylation of serine, threonine, and tyrosine residues, are crucial for regulating protein interactions[14,15]. Src, a non-receptor tyrosine kinase, plays a central role in cellular signaling networks, influencing various physiological processes[16]. Reduction of Src expression in cell monolayers mitigates lipopolysaccharide-induced increases in permeability, ZO-1 redistribution, and Occludin expression[17]. Overexpression of a kinase-inactive mutant Src alleviates tight junction damage induced by oxidative stress and expedites $Ca^{2+}$-mediated tight junction reorganization in Caco-2 cells[18]. Preincubation of MDCK kidney cells with PP2, an inhibitor of Src kinase, increases the permeability of tight junctions in intestinal epithelial cells compared to the control group[19]. Although the results of these studies were not entirely consistent across different treatments, they collectively highlight the critical role of Src kinase in regulating tight junction function. As a ubiquitously expressed tyrosine protein kinase, Src has been reported to phosphorylate tight junction proteins or associated proteins[20–22]. Still, there are few reports on the functional relevance of these Src phosphorylation sites in tight junctions.

PEAK1 (Pseudopodium-Enriched Atypical Kinase 1) is a pseudokinase implicated in cell signaling and structural organization[23–27]. It binds to F-actin and localizes at focal adhesions, influencing processes such as adhesion and migration[27,28]. PEAK1 is known to carry extensive tyrosine phosphorylated sites[29], some of which have proven functional relevance. For example, dynamic regulation of PEAK1's Tyr-665 phosphorylation modulates the assembly and disassembly of cell focal adhesions[30], while mutation of its Tyr-635 to phenylalanine reduces acinar size and cellular invasion in basal breast cancer cells[31]. Although the function of many phosphorylated-tyrosine sites on PEAK1 is unknown, Src is a well-established PEAK1 kinase in cancer cells[25,30,31]. While PEAK1's roles in cancer cells are well documented, its regulatory mechanisms and physiological functions in non-cancerous contexts require further exploration. Analyzing a patient dataset (GDS3119)[32] revealed decreased PEAK1 expression in the colonic epithelia of patients with active ulcerative colitis compared to non-inflamed controls (Supplementary Fig. 1a), suggesting that PEAK1 could be related to the progression of ulcerative colitis. Notably, the role of PEAK1 in enteritis has not yet been described.

In this study, we demonstrate that *Peak1⁻/⁻* mice exhibit increased intestinal tight junction-dependent permeability and enhanced vulnerability to DSS-induced colitis. Lack of PEAK1 disrupts barrier integrity in intestinal epithelial cells, affecting tight junction structure and electrical resistance in Caco-2 cells. Mechanistically, PEAK1 localizes at tight junctions, interacting with ZO-1 through a specific amino acid region (714-731) that includes a crucial Src-phosphorylated residue (Y724). PEAK1 deficiency or disruption of PEAK1/ZO-1 interaction exposes the LC3-interacting region on ZO-1, triggering autophagy-mediated ZO-1 degradation and disrupting cellular connections. Additionally, PEAK1 deficiency is associated with increased CSK-Src interaction and a concomitant decrease in Src activity in intestinal epithelial cells. Our functional and molecular findings conclusively demonstrate that the PEAK1/CSK/Src complex is crucial for maintaining tight junctions, highlighting a vital role for PEAK1 in preserving intestinal barrier integrity.

## Results

### PEAK1 deficiency disrupts tight junction structure in intestinal epithelial cells

To investigate a possible involvement of PEAK1 in intestinal inflammation, we induced colitis by administering mice with 2% Dextran Sulfate Sodium (DSS) for 7 days, followed by switching to sterilized water. C57BL/6 mice exhibited a decline in body weight starting on day 6, reaching the lowest point on day 11, and then gradually recovering (Supplementary Fig. 1b); a similar pattern was observed in the colon length of these mice (Supplementary Fig. 1c). Immunoblot analysis revealed that the levels of ZO-1, Occludin, and Claudin 2 decreased to their lowest points and then gradually recovered, paralleling changes in body weight and colon length (Fig. 1a). Interestingly, PEAK1 exhibited a similar expression pattern to ZO-1 in DSS-treated mice (Fig. 1a). Hematoxylin-Eosin (H&E) staining and immunohistochemistry indicated a negative correlation between PEAK1 expression and the intensity of colon inflammation (Fig. 1b), suggesting PEAK1's involvement in regulating the intestinal barrier. To explore this, we generated PEAK1 knockout mice (*Peak1⁻/⁻*) using CRISPR/Cas9 (Supplementary Fig. 1d, e). Notably, upon PEAK1, loss the levels of ZO-1, Occludin, and Claudin 2 significantly decreased in colon tissue and intestinal epithelial cells (IECs) isolated from the colon of these mice (Fig. 1c, d), without changes in their mRNA levels (Supplementary Fig. 1f–m). In addition, PEAK1 deficiency reduced ZO-1 and Claudin 2 distribution at the apical region of intestinal epithelial cells (Fig. 1e; Supplementary Fig. 1n–q) without disordered microvilli in the colon (Supplementary Fig. 1r). Given that downregulation of ZO-1 is known to augment Claudin 2 degradation[33], we focus our efforts on addressing the molecular mechanism by which PEAK1 regulates ZO-1 protein levels. To further investigate the link between PEAK1 and ZO-1, we created PEAK1 CRISPR-knockout Caco-2 cells, a colon carcinoma cell commonly used for studying tight junctions. In these PEAK1-depleted epithelial cells, protein levels of ZO-1 and Occludin were reduced (Fig. 1f; Supplementary Fig. 2a). This was confirmed by knocking down PEAK1 using small interfering RNA in different colon-derived cell lines (Supplementary Fig. 2b, c). Like our findings in mice, the reduction of ZO-1 and Occludin protein levels in PEAK1-depleted cells occurred independently of mRNA changes (Supplementary Fig. 2d–g). The binding between ZO1 and Occludin, essential for tight junction homeostasis[34,35], was noticeably increased following PEAK1 depletion (Supplementary Fig. 2h, i). Morphological analysis using a computational model[36] (Fig. 1g, h; Supplementary Fig. 2j), showed increased tortuosity of tight junctions in Caco-2 cells after PEAK1 loss (Fig. 1i; Supplementary Fig. 2k). Transmission electron microscopy of colon cross-sections from *Peak1⁻/⁻* mice revealed a reduced cell membrane fusion between two adjacent intestinal epithelial cells (Fig. 1j; Supplementary Fig. 2l). *Peak1⁻/⁻* mice had increased intestinal permeability to 4 kDa FITC-dextran and 113 Da creatinine (Fig. 1k, l), but not the larger 70 kDa Rhodamine-dextran (Fig. 1m), indicating defects in tight junction-dependent permeability rather than unrestricted permeability. In Caco-2 cells, deletion of PEAK1 resulted in decreased transepithelial electrical resistance (TEER) and higher permeability to FITC-dextran (Fig. 1n, o), indicating loss of barrier integrity. Overexpression of ZO-1 in PEAK1 knockout cells restored the disrupted tight junction organization, the reduction in TEER, and the enhanced permeability to FITC-dextran caused by PEAK1 deficiency (Supplementary Fig. 2m–q). Thus, PEAK1 deletion reduces the levels of ZO-1 and Occludin, disrupting tight junction structure and function both in vivo and in vitro.

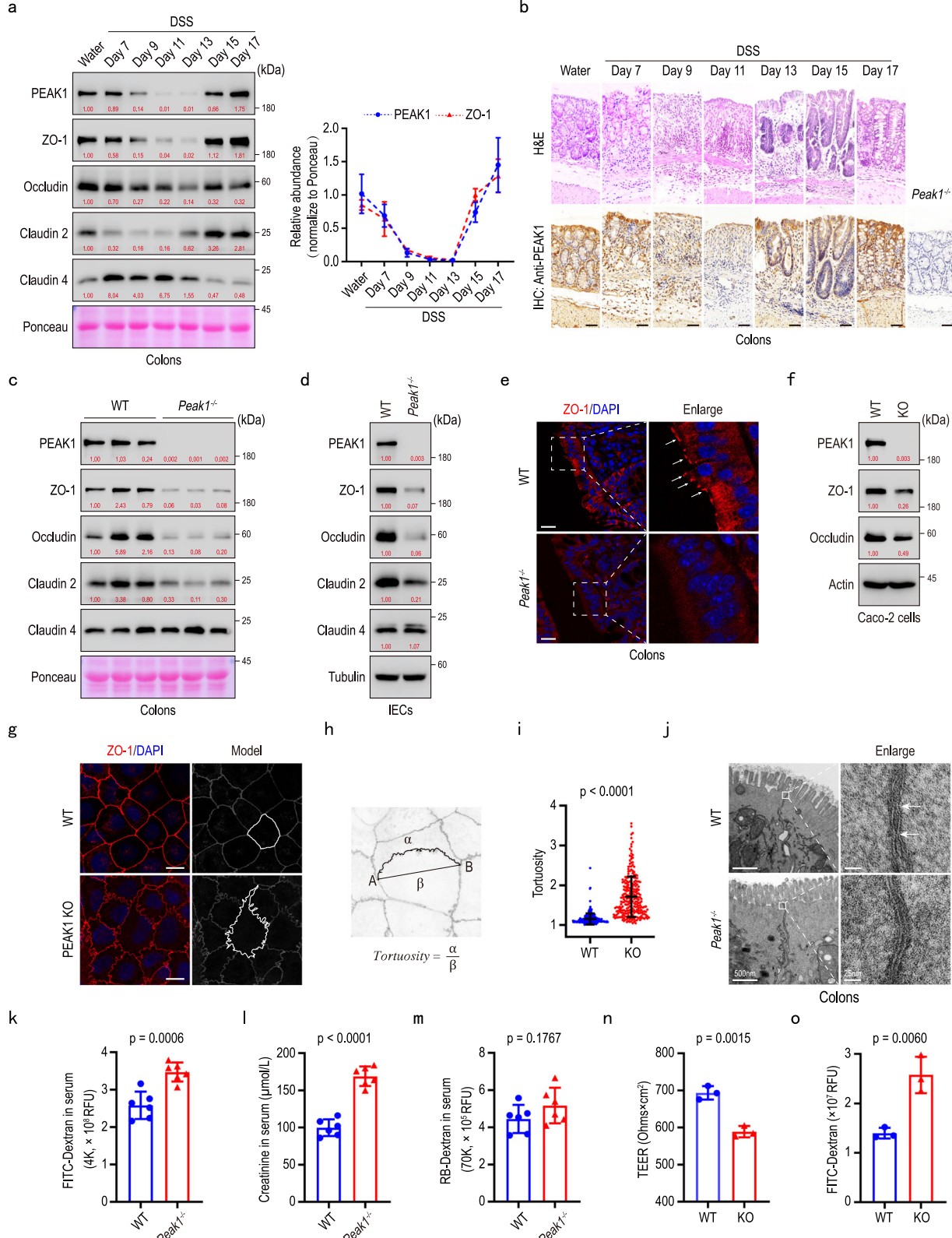

### Y724 on PEAK1 modulates its interaction with ZO-1

Given the significant role of protein-protein interactions in orchestrating tight junctions and our findings that PEAK1 controls ZO-1 and Occludin at the protein level, we investigated whether PEAK1 interacts with tight junction proteins. To test this, we overexpressed GFP-PEAK1 in HEK293T cells and performed immunoprecipitation (Supplementary Fig. 3a). Mass spectrometry identified ZO-1 and Occludin as

potential interaction partners of PEAK1 (Supplementary Fig. 3b). This was confirmed by co-immunoprecipitating endogenous PEAK1 and ZO-1 in Caco-2 cells (Fig. 2a; Supplementary Fig. 3c). In epithelial cell lines, PEAK1 was primarily localized at cell-cell contact sites, where it co-localized with ZO-1 (Fig. 2b, c; Supplementary Fig. 3d). To further understand the PEAK1/ZO-1 interaction, we mapped the interaction sites using a series of truncated PEAK1 constructs,

**Fig. 1 | PEAK1 deficiency disrupts the structure of the intestinal epithelial cell tight junction. a** Representative western blot of PEAK1, ZO1, Occludin, Claudin 2 and Claudin 4 (left) as well as relative quantification (right) of PEAK1 and ZO-1 levels in colons of WT mice challenged with a 2% DSS-induced colitis model over 17 days (day 0-8: 2% DSS, day 8-17: H$_2$O). Ponceau staining was used as a loading control. Data are shown as mean ± SD for three independent experiments. **b** H&E staining for morphology and inflammatory cell infiltration analysis (top), and immunohistochemistry for detection of PEAK1 expression (bottom) in colon cross-sections from WT mice treated with 2% DSS at the different time points. Scale bars, 50 μm. Images from sections of *Peak1* knockout (*Peak1$^{-/-}$*) mice, showing no staining, validate the specificity of the anti-PEAK1 antibody. **c–d** Representative western blot showing PEAK1, ZO-1, Occludin, Claudin 2, and Claudin 4 levels in the colon tissues (**c**) and isolated colonic epithelial cells (IECs, **d**) from WT and *Peak1$^{-/-}$* mice. Claudin 4 and beta-tubulin were used as loading controls, respectively.
**e** Immunofluorescence for ZO-1 (red) in colon cross-sections from WT and *Peak1$^{-/-}$* mice. The dotted box indicates the area that is enlarged in the right images. White arrows indicate the regions with the highest ZO-1 high expression. Scale bars, 20 μm. **f** Western blot demonstrating efficient PEAK1 knockout (KO) in Caco-2 cells, and a concomitant decrease in ZO-1 and Occludin levels in KO compared to wild-type (WT) controls, using beta-actin as a loading control. **g** Representative

immunofluorescence images showing ZO-1 (red) expression in WT and PEAK1 KO Caco-2 cells. DAPI was used for nuclear counterstaining (blue). Scale bars, 15 μm. **h** Schematic diagram and mathematical formula used for assessing the tortuosity of cell tight junctions. **i** Quantification of barrier tortuosity (as assessed in h). Data are presented as mean ± SD. WT, *n* = 261 cells; KO, *n* = 260 cells. Unpaired two-tailed Mann-Whitney test. **j** Transmission electron microscopy (TEM) images showing tight junctions in colon cross-sections from WT and *Peak1$^{-/-}$* mice. White boxes in the left images indicate the regions that are enlarged in the right images, highlighting gaps between tight junctions. Left panel scale bars, 500 nm. Right panel scale bars, 25 nm. **k–m** Serum levels of 4 kDa FITC-dextran (**k**), creatinine (**l**), and 70 kDa Rhodamine B (RB)-dextran (**m**) in WT and *Peak1$^{-/-}$* mice. Data are presented as mean ± SD. WT, *n* = 6 mice; *Peak1$^{-/-}$*, *n* = 6 mice. Unpaired two-tailed Student's t-test. **n–o** Transepithelial electrical resistance (TEER, **n**), and fluorescence intensity of FITC-dextran in the lower chambers of trans-well inserts (0.4 μm) seeded with WT or PEAK1 KO Caco-2 cell monolayers (**o**). Data are expressed as mean ± SD for three biological replicates. Unpaired two-tailed Student's t-test. All experiments were repeated three times, yielding similar results. The quantified relative expression levels shown in western blots are indicated in red, and represents the results of at least three repeated experiments.

according to predicted domains[28] (Supplementary Fig. 3e). Our mapping results indicated that the PEAK1 fragment spanning amino acids 700 to 750 interacts with ZO-1 (Fig. 2d, e), with residues 714-731 being highly conserved across different species (Supplementary Fig. 3f). Deleting this fragment from the full-length PEAK1 (Δ714-731) (Supplementary Fig. 3g) substantially attenuated PEAK1's binding to ZO-1 (Fig. 2f), suggesting a critical role for this segment in mediating the PEAK1/ZO-1 interaction.

Since phosphorylation of amino acid residues dynamically regulates protein localization, interactions, and degradation, we sequentially mutated all serine (S) in the identified PEAK1 conserved segment to alanine (A) (Supplementary Fig. 3h). None of the mutations impaired the interaction between PEAK1 and ZO-1 (Supplementary Fig. 3i), indicating that serine phosphorylation of this fragment is not relevant. Further mapping for the essential binding region pinpointed PEAK1's residues 720-726 as essential for binding ZO-1 (Fig. 2g; Supplementary Fig. 3j). Strikingly, mutating to phenylalanine (F) the single tyrosine (Y) residue (Y724F) located within the 720-726 region markedly impaired PEAK1-ZO-1 interaction (Supplementary Fig. 3k; Fig. 2h, i). In summary, tyrosine Y724 on PEAK1, located within the core conserved fragment 714-731, plays a pivotal role in modulating its interaction with ZO-1.

## Src-phosphorylated PEAK1 Y724 contributes to maintaining tight junction integrity

To determine whether PEAK1's Y724 tyrosine undergoes phosphorylation and affects tight junction integrity in cells, we overexpressed and immunoprecipitated GFP-tagged wild-type (WT) and Y724F mutant PEAK1 in HEK293T cells. Immunoblotting with the 4G10 antibody, which recognizes phosphorylated tyrosine residues, showed decreased total tyrosine phosphorylation in Y724F point mutant PEAK1 compared to WT PEAK1 (Fig. 3a). LC-MS/MS identified both phosphorylated and unphosphorylated Y724-containing peptides on GFP-PEAK1 (Fig. 3b; Supplementary Fig. 4a). We generated an antibody specific to phosphorylated Y724, validated by immunoblotting of WT and Y724F PEAK1, and confirmed that Y724 is phosphorylated in cells (Supplementary Fig. 4b). To identify the kinase responsible for phosphorylating PEAK1 at Y724, we treated Caco-2 cells with various kinase inhibitors. Only PP2, a Src family kinase (SFK) inhibitor, significantly diminished Y724 phosphorylation, suggesting that Y724 is regulated by SFKs (Fig. 3c, d; Supplementary Fig. 4c).

Src, a prominent SFK[37] and known tyrosine kinase, has been previously identified as the kinase responsible for phosphorylating PEAK1 on Y635[31] and Y665[30]. Src also appeared in our list of potential interacting proteins for PEAK1 (Supplementary Fig. 3b). To validate Src as

the kinase regulating Y724 phosphorylation, we used anti-pY724 (PEAK1) immunoblotting to detect the phosphorylation of Y724 on WT and Y724F mutant PEAK1 after Src overexpression. Whereas Src significantly increased the phosphorylation of PEAK1 Y724, no bands were observed on lysates from cells expressing the PEAK1 Y724F mutant, regardless of Src expression (Fig. 3e). Moreover, we constructed a kinase-inactive mutant of Src (K298M)[38] (Supplementary Fig. 4d). In vitro kinase assays using purified GST-Src$^{WT}$ or GST-Src$^{K298M}$ in combination with FC-PEAK1$^{WT}$ or FC-PEAK1$^{Y724F}$ proteins showed that only Src$^{WT}$, but not the kinase-dead Src$^{K298M}$, could enhance PEAK1 Y724 phosphorylation (Fig. 3f, Supplementary Fig. 4e, f). Together, these results confirm that Y724 on PEAK1 is a substrate site for Src kinase.

Given the central role of Y724 in regulating PEAK1/ZO-1 interaction (Fig. 2h, i), we speculated that phosphorylation of PEAK1 Y724 is key to maintaining tight junction integrity. To assess the functional role of Y724 phosphorylation, we mutated PEAK1 Y724 to F in Caco-2 cells using CRISPR-Cas9 (Supplementary Fig. 4g, h). Compared to WT cells, PEAK1 Y724F-expressing Caco-2 cells exhibited reduced ZO-1 expression (Fig. 3g). The phosphorylated Y724 form of PEAK1, but not its non-phosphorylated mutant (Y724F), was able to bind to ZO-1 (Fig. 3h). The mutation also caused increased tight junction tortuosity (Fig. 3i, j), decreased transepithelial electrical resistance (TEER), and higher FITC-dextran permeability of monolayer cells (Fig. 3k, l). Importantly, in PEAK1-depleted Caco-2 cells, overexpression of PEAK1 Y724F failed to restore the structural and functional integrity of tight junctions lost due to PEAK1 deficiency. This included persistent tight junction tortuosity (Supplementary Fig. 4i–k), reduced TEER (Supplementary Fig. 4l), increased FITC-dextran permeability in monolayer cells (Supplementary Fig. 4m), and decreased levels of ZO-1 and Occludin (Supplementary Fig. 4n). Thus, phosphorylation of PEAK1 at Y724 by Src is crucial for maintaining tight junction integrity.

## PEAK1 deficiency decreases Src activity and disrupts tight junctions

The phosphorylation of Src kinase at Y416 and Y527 represents its active and inactive kinase forms, respectively[37]. PEAK1 interacted with both active and inactive Src forms (Fig. 4a). In different intestinal epithelial cell lines as well as isolated murine intestinal colonic epithelium, PEAK1 deletion markedly reduced Src Y416 phosphorylation and increased Y527 phosphorylation, indicating diminished Src kinase activity in PEAK1's absence (Fig. 4b, c; Supplementary Fig. 5a, b). In Caco-2 cells, inhibiting Src activity with PP2 decreased PEAK1/ZO-1 and PEAK1/Src interactions, reduced ZO-1 expression, and increased Occludin-ZO-1 binding (Fig. 4d; Supplementary Fig. 5c).

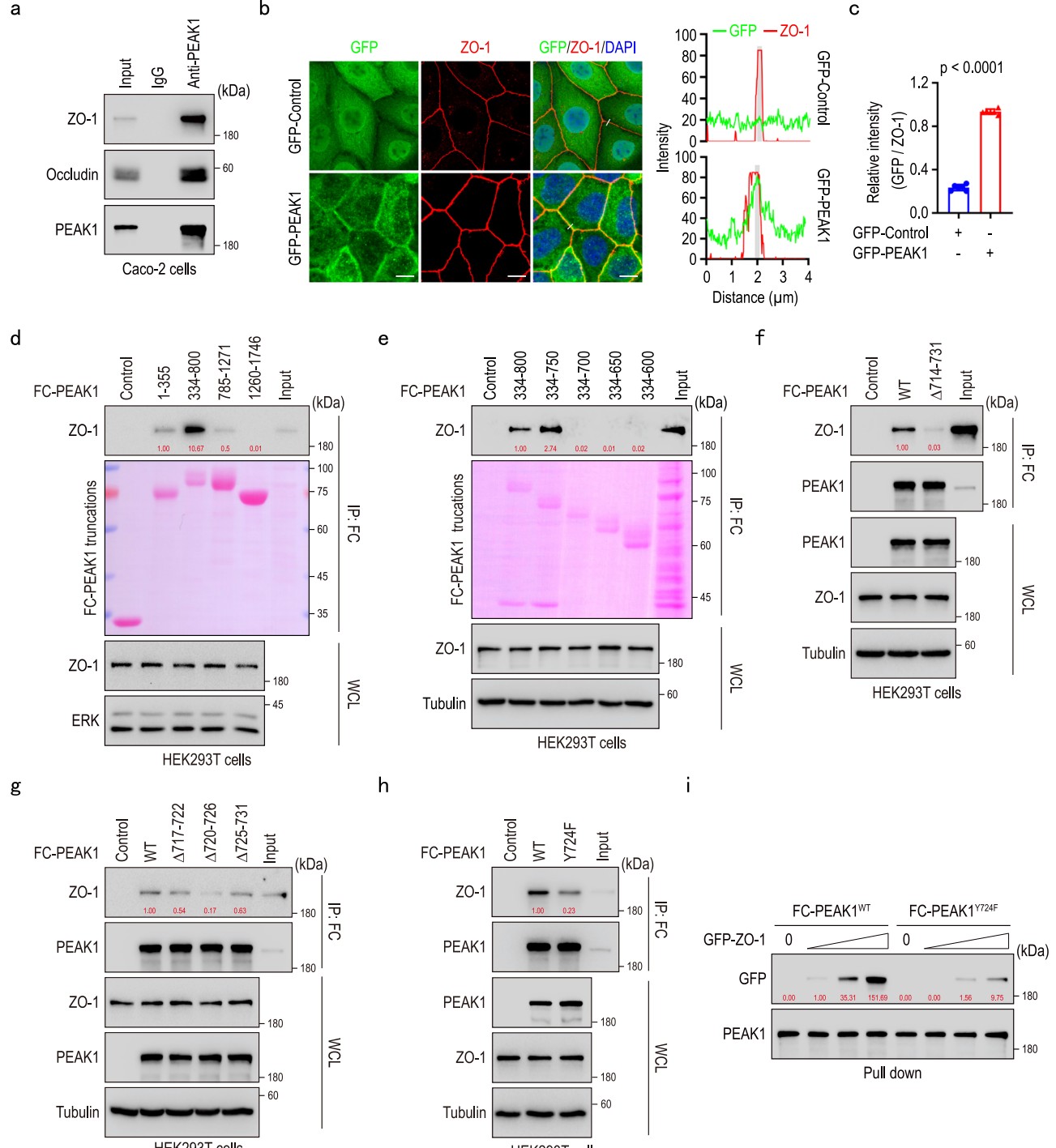

**Fig. 2 | Y724 on PEAK1 modulates its interaction with ZO-1. a** Co-immunoprecipitation of endogenous PEAK1 from Caco-2 cell protein lysates using anti-PEAK1 antibody, revealing interaction between PEAK1, ZO-1, and Occludin. **b** Representative immunofluorescence images showing GFP-PEAK1 localization in Caco-2 cells, with GFP-PEAK1 (green), ZO-1 (red), and DAPI (nuclei, blue) staining (left panel). The right panel shows the co-localization analysis of GFP and ZO-1. Areas used for fluorescence intensity quantification are colored in gray. Scale bars, 10 μm. **c** Relative quantification of GFP and ZO-1 fluorescence intensity from the gray areas shown in (b). Data are presented as mean ± SD. *n* = 6 independent samples. Unpaired two-tailed Student's t-test. **d–e** Co-immunoprecipitation assays to map PEAK1's binding regions to ZO-1, using various overexpressed FC-tagged

PEAK1 truncated proteins. Ponceau staining was used to visualize the truncated proteins. **f–h** Co-immunoprecipitation of HEK293T cell lysates overexpressing FC-Control (Control), FC-tagged PEAK1$^{WT}$ (WT), and various PEAK1 mutants to evaluate the impact of specific PEAK1 truncations on the PEAK1/ZO-1 interaction. **i** Pull-down assays for the interaction of PEAK1 and ZO-1, and its dependence on the PEAK1 Y724 site. FC-tagged PEAK1$^{WT}$ and FC-tagged PEAK1$^{Y724F}$ proteins, purified from HEK293T cells, were incubated with GFP-tagged ZO-1 protein at increasing concentrations (0, 1, 2, and 4 μg) to assess their binding. The quantified relative expression levels shown in western blots are indicated in red, and represents the results of at least three repeated experiments.

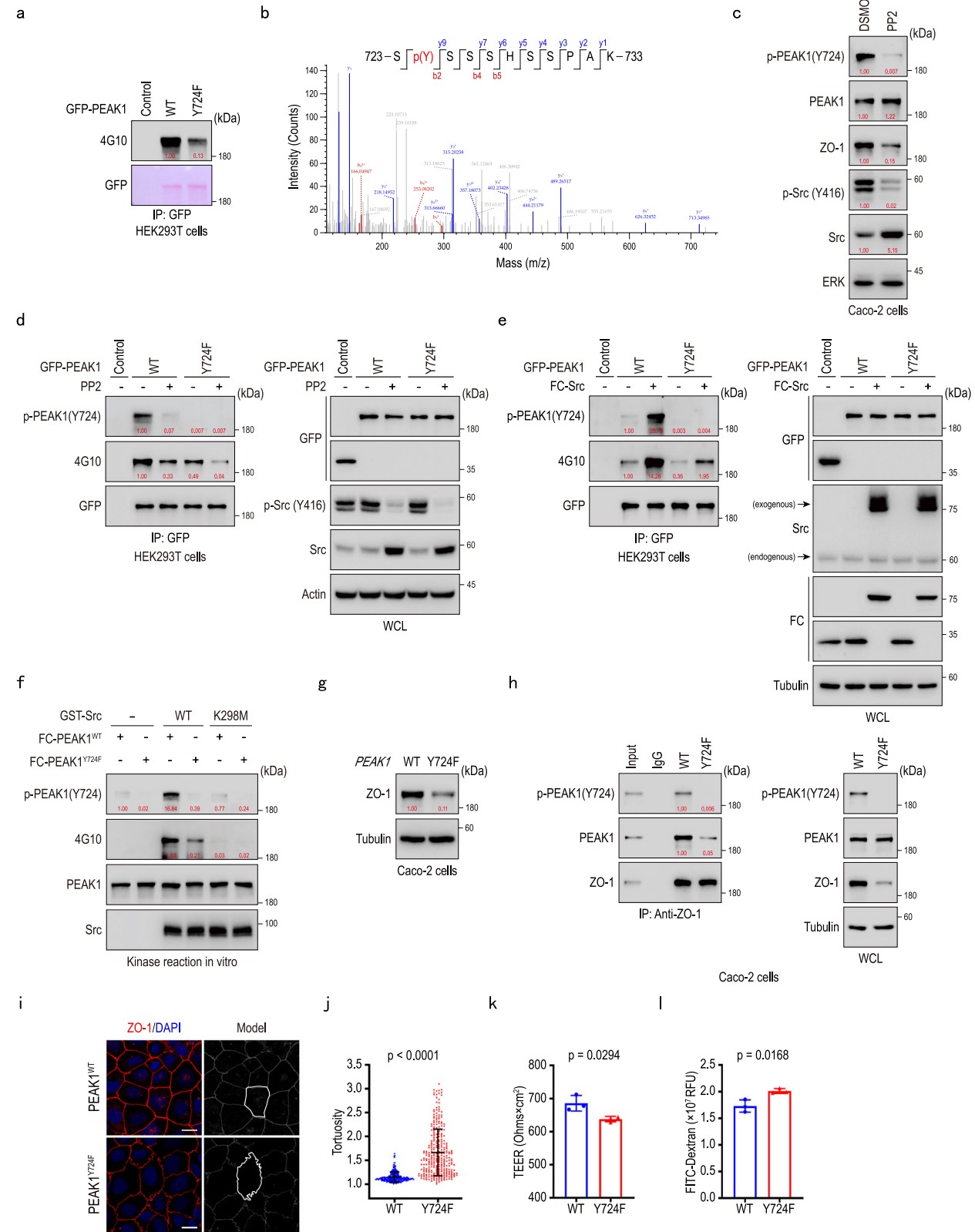

Morphologically, similar to our observations in PEAK1-deficient cells (Fig. 1g; Supplementary Fig. 2j), PP2 treatment led to abnormal tight junctions, characterized by petal-like distortions and tortuosity (Fig. 4e, f), lower TEER (Fig. 4g), and higher FITC-Dextran permeability (Fig. 4h), underscoring the critical role of Src kinase activity in tight junction integrity.

To further elucidate the role of Src activity in tight junction maintenance, we constructed kinase-active (Y530F) and -inactive Src (K298M) mutants[38] (Supplementary Fig. 5d; Supplementary Fig. 4d). Overexpression of Src[WT] or constitutively active Src[Y530F] increased ZO-1 levels, augmented TEER, and decreased FITC-Dextran permeability, indicating enhanced tight junction formation and support of barrier

**Fig. 3 | Src-phosphorylated PEAK1 Y724 contributes to maintaining the integrity of tight junctions. a** Immunoprecipitation of HEK293T cell lysates over-expressing GFP-Control (Control), GFP-tagged PEAK1$^{WT}$ (WT), and GFP-tagged PEAK1$^{Y724F}$ mutant (Y724F) plasmids, to detect PEAK1 tyrosine phosphorylation using an anti-phosphotyrosine antibody (4G10). GFP was visualized by Ponceau staining. **b** Identification of PEAK1 Y724 phosphorylation by mass spectrometry. **c** Representative western blot of Caco-2 cell lysates treated with 10 μM PP2 inhibitor. **d** PEAK1 immunoprecipitation from HEK293T cell lysates overexpressing GFP-tagged PEAK1$^{WT}$ or GFP-tagged PEAK1$^{Y724F}$, to determine PEAK1 phosphorylation following treatment with 10 μM PP2 for 2 h. **e** Immunoprecipitation of GFP-tagged PEAK1 from HEK293T cells co-overexpressing GFP-tagged PEAK1$^{WT}$ or GFP-tagged PEAK1$^{Y724F}$ with FC-tagged Src. **f** In vitro kinase assay using different FC-tagged PEAK1 and GST-tagged Src proteins in the presence of ATP.

**g** Representative western blot showing ZO-1 levels in Wild-type (WT) and PEAK1$^{Y724F}$ mutant Caco-2 cell lysates. **h** Co-immunoprecipitation of endogenous ZO-1 from WT and PEAK1$^{Y724F}$ mutant Caco-2 cell lysates using anti-ZO-1 antibody. **i, j** Representative immunofluorescence images (**i**) and barrier tortuosity quantification (**j**) in PEAK1$^{WT}$ and PEAK1$^{Y724F}$ Caco-2 cells. Scale bars, 15 μm. Data are presented as mean ± SD. $n = 270$ cells for PEAK1$^{WT}$; $n = 299$ cells for PEAK1$^{Y724F}$. Unpaired two-tailed Mann-Whitney test. **k, l** TEER (**k**) and fluorescence intensity of FITC-dextran (**l**) in the lower chambers of trans-well inserts (0.4 μm) seeded with PEAK1$^{WT}$ or PEAK1$^{Y724F}$ Caco-2 monolayers. Data are Mean ± SD. $n = 3$ biological replicates per group. Unpaired two-tailed Student's t-test. The quantified relative expression levels shown in western blots are indicated in red, and represents the results of at least three repeated experiments.

function (Fig. 4i–m; Supplementary Fig. 5e, f). Cells expressing the kinase-dead Src$^{K298M}$ mutant exhibited no significant changes in ZO-1 levels or TEER, confirming that Src's catalytic kinase activity is essential for tight junction regulation (Supplementary Fig. 5e, f). Importantly, in Caco-2 cells expressing the PEAK1 Y724F mutant, overexpression of Src kinase was insufficient to reverse the detrimental effects of the PEAK1 mutation, as evidenced by the continued presence of decreased ZO-1 levels, lower TEER, increased tight junction distortions, and FITC-Dextran permeability (Fig. 4i–m), regardless of Src levels. These findings reveal that Src kinase actions on maintaining tight junctions rely on the functional role of PEAK1 Y724. In summary, PEAK1 supports Src kinase activity, and Src, in turn, increases tyrosine phosphorylation at the PEAK1 Y724 site to maintain tight junction barrier function.

## PEAK1-CSK binding regulates Src activity and tight junction integrity

We demonstrated that PEAK1 is a phosphorylation substrate of Src (Fig. 3e, f), and at the same time, we observed decreased Src activity in cells and murine colons lacking PEAK1. To elucidate the underlying regulatory mechanism, we re-examined the list of PEAK1 interaction partners (Supplementary Fig. 3b) and confirmed via immunoprecipitation that CSK, a known inhibitor of Src kinase activity[39], interacts with PEAK1 (Fig. 5a). The increased Src Y527 phosphorylation observed in PEAK1-depleted cells also suggested CSK's involvement (Fig. 4b, c; Supplementary Fig. 5a, b). Immunoprecipitation with full-length and truncated PEAK1 constructs revealed that the 861-879 amino acid region of PEAK1 is responsible for interacting with CSK (Fig. 5a, b; Supplementary Fig. 6a, b). To investigate the interplay among PEAK1, CSK, and Src, we analyzed Caco-2 cells either lacking or stably expressing CSK. CSK knockout reduced PEAK1's binding to Src (Fig. 5c), and PEAK1 depletion promoted CSK-Src interaction (Fig. 5d; Supplementary Fig. 6c, d). CSK overexpression increased PEAK1's binding to Src (Fig. 5e), whereas inhibition of Src kinase activity with PP2 did not affect PEAK1's binding to CSK (Supplementary Fig. 6e). These findings indicate that CSK effectively enhances the interaction between PEAK1 and Src, while PEAK1 attenuates the binding of CSK to Src to sustain Src kinase activity.

Given the role of CSK in the PEAK1-Src complex, we explore CSK's contribution to tight junction integrity. Overexpression of CSK led to decreased TEER values (Supplementary Fig. 6f, g), indicating that CSK weakens tight junctions. In contrast, CSK knockout in Caco-2 cells increased phosphorylation of PEAK1 at the Y724 site and enhanced ZO-1 levels (Fig. 5f), with no overt morphological differences in tight junction organization (Fig. 5g, h). Importantly, in Caco-2 cells carrying the Y724F PEAK1 mutation, CSK knockout failed to restore the structural and functional impairments caused by the mutation, including reduced ZO1 expression, lower TEER, increased tight junction tortuosity, and FITC-Dextran permeability (Fig. 5f–j). Thus, similar to Src, CSK's effects on tight junctions depend on the functional role of PEAK1 Y724. In summary, PEAK1 interacts with CSK to sustain Src activation and, in turn, its Y724 phosphorylation.

## PEAK1 deletion triggers ZO-1 degradation by exposing its LC3-interacting region

Understanding how PEAK1 deficiency leads to reduced ZO-1 is crucial for revealing PEAK1's role in tight junction integrity. We first treated PEAK1-depleted Caco-2 cells with the proteasome inhibitor MG132, but ZO-1 levels were not restored (Supplementary Fig. 7a). In contrast, treatment with the autophagy inhibitor Bafilomycin A1 (Baf A1) did rescue ZO-1 levels in PEAK1-knockout Caco-2 cells (Fig. 6a; Supplementary Fig. 7b). Conversely, autophagy activation with rapamycin (Rapa) promoted ZO-1 degradation in various cells (Fig. 6b; Supplementary Fig. 7c, d). Additionally, ZO-1 interacted and spatially co-localized with LC-3B in Caco-2 cells (Fig. 6c, d). These data indicate that in the absence of PEAK1, ZO-1 is degraded by autophagy.

To molecularly understand how PEAK1 deficiency promotes ZO-1 degradation, we mapped the domain of ZO-1 that binds to PEAK1 and found that the interaction occurs through amino acids 1151 to 1371 (Fig. 6e; Supplementary Fig. 7e). Computational predictions identified four LC3-interacting regions (LIR) within full-length ZO-1, with two located in the identified ZO-1/PEAK1 binding region (1151-1371 amino acids) (Fig. 6f). By constructing full-length deletions, we revealed that ZO-1 mutants lacking the segment between amino acids 1212 and 1217 (Δ1212-1217) had the weakest interaction with LC-3B (Supplementary Fig. 7f–h) and PEAK1 (Fig. 6g, h), suggesting that both PEAK1 and LC-3B bind to this motif. Furthermore, in PEAK1-deficient Caco-2 cells, complementation with both ZO-1$^{FL}$ and ZO-1$^{Δ1212-1217}$ proteins restored the disorganized tight junction arrangement, the decreased TEER, and the increased permeability to FITC-Dextran caused by PEAK1 deficiency (Supplementary Fig. 7i–m).

Our previous results showed that PEAK1 loss, the PEAK Y724F mutation, or Src inhibition with PP2 all reduce ZO-1 levels and disrupt tight junction function (Figs. 1, 3, and 4). We hypothesized that the underlying mechanism involves reduced phosphorylation at the PEAK1 Y724 site, which increases the binding of ZO-1 to LC-3B, thereby promoting ZO-1 degradation via autophagy. We confirmed this hypothesis with co-immunoprecipitation and immunofluorescence analyses (Fig. 6i–n). CSK deficiency enhanced Src activity and reduced the binding of ZO-1 to LC-3B. However, CSK knockout cannot fully counteract the increased ZO-1/LC-3B interaction induced by the Y724F PEAK1 mutation (Fig. 6m), confirming that the effects of CSK are dependent on PEAK1.

In summary, under physiological conditions, PEAK1 competes with LC-3B for binding to the LIR motif of ZO-1. Reduced phosphorylation at the PEAK1 Y724 site weakens the interaction between PEAK1 and ZO-1, facilitating LC-3B binding to the LIR motif of ZO-1 and leading to ZO-1 degradation via the autophagic pathway.

## PEAK1 deletion increases susceptibility to DSS-induced colitis

The intestinal barrier, as the first line of defense against the invasion of pathogenic microorganisms, plays a vital role in maintaining body homeostasis[40,41]. Although Peak1$^{-/-}$ mice did not develop spontaneous intestinal inflammation (Supplementary Fig. 8a–d), they displayed

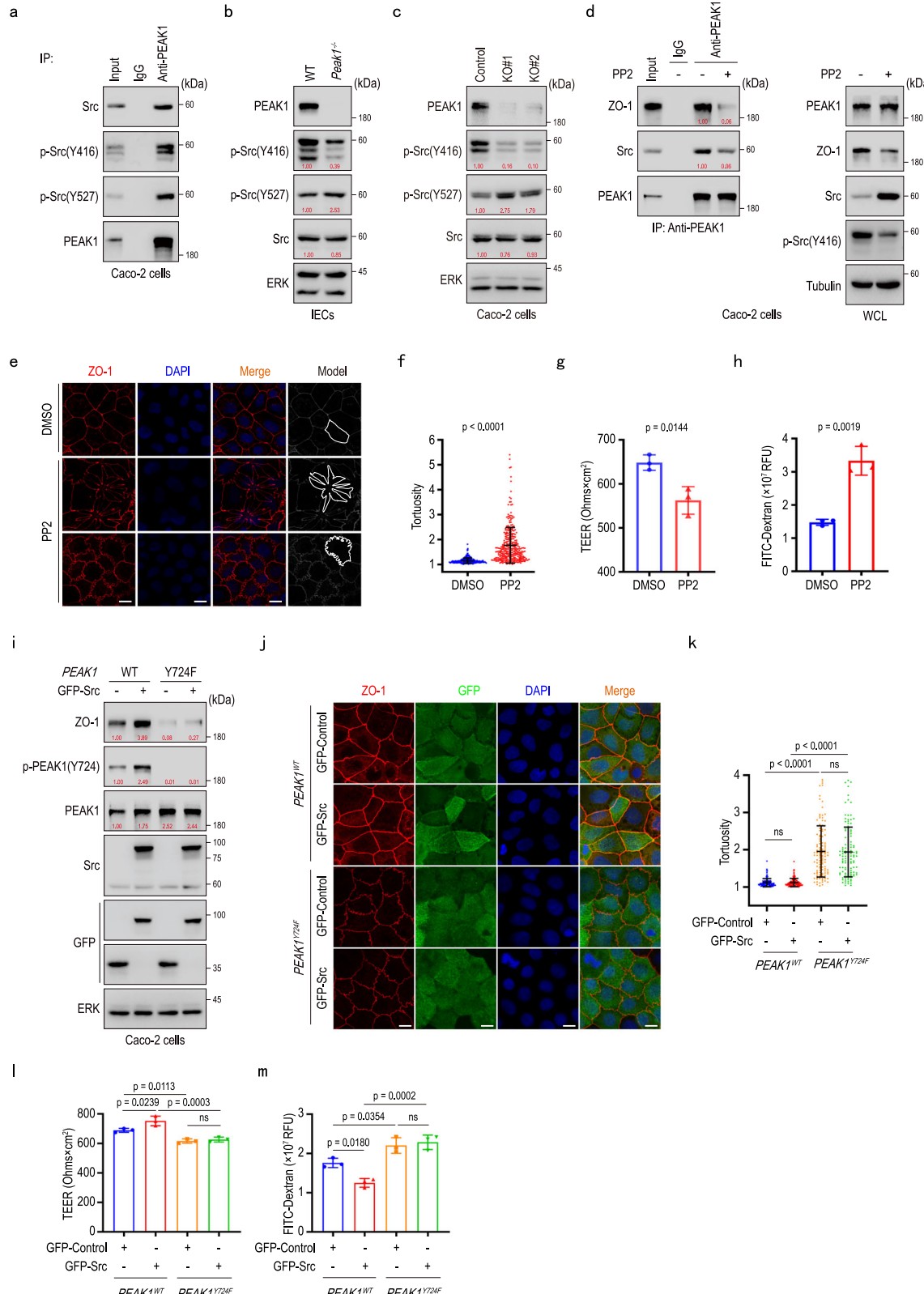

worsened symptoms following colitis induction with 2% DSS compared to WT mice. This was particularly evident during the repair phase, as *Peak1*[−/−] mice displayed slower body weight repair (Fig. 7a), higher mortality (Fig. 7b), shorter colons (Fig. 7c, d), fewer Ki67-positive cells (Supplementary Fig. 8e), and increased TUNEL-positive cells in colons (Supplementary Fig. 8f). Thus, PEAK1 deficiency, like ZO-1 deficiency[12], impairs intestinal repair following DSS-induced damage.

Histopathological analysis via H&E staining revealed greater immune cell infiltration, thicker colonic walls, more severe crypt disruption, and enlarged gaps of tight junctions in DSS-treated *Peak1*[−/−] mice (Fig. 7e–g; Supplementary Fig. 8g). Furthermore, these *Peak1*[−/−] mice exhibited a significant increase in F4/80-positive infiltrating macrophages, along with higher levels of pro-inflammatory factors TNF-α and IL-6, and lower levels of the anti-inflammatory factor IL-10

**Fig. 4 | PEAK1 deficiency decreases Src activity and disrupts tight junction integrity. a** Immunoprecipitation of Caco-2 cell lysates with an anti-PEAK1 antibody to detect PEAK1 binding to Src, p-Src (Y416, active) and p-Src (Y527, inactive) forms. **b, c** Western blot for detecting the levels of active (p-Src Y416) and inactive (p-Src Y527) forms in isolated intestinal epithelial cells (IECs) from WT and *Peak1⁻ᐟ⁻* mice (**b**), as well as WT and PEAK1 knockout Caco-2 cells (**c**). **d** Co-immunoprecipitation in Caco-2 cell lysates using anti-PEAK1 antibodies to detect binding of PEAK1 to ZO-1 and Src upon treatment with 10 µM of the Src inhibitor PP2 for 2 h. **e, f** Representative immunofluorescence images (e) and barrier tortuosity quantification (f) in Caco-2 cells treated with 10 µM PP2 inhibitor or DMSO (control) for 2 h. Scale bars, 20 µm. Data are presented as mean ± SD. DMSO, *n* = 271 cells; PP2, *n* = 314 cells. Unpaired two-tailed Mann-Whitney test. **g, h** Transepithelial electrical resistance (TEER) (g) and fluorescence intensity of FITC-dextran (h) in the lower chambers of trans-well inserts (0.4 µm) seeded with Caco-2 cells treated with DMSO (controls)

or 10 µM PP2 inhibitor for 2 h. Data are presented as mean ± SD for three biological replicates. Unpaired two-tailed Student's t-test. **i** Western blot analysis in WT and PEAK1 Y724F mutant Caco-2 cells overexpressing GFP-tagged Src. **j, k** Representative confocal images (**j**) and barrier tortuosity quantification (**k**) in WT and PEAK1 Y724F Caco-2 cells transfected with GFP-tagged Src or GFP-Control plasmids. Scale bars, 10 µm. Data are shown as mean ± SD. *n* = 118 cells per group. Kruskal-Wallis test, followed by Dunn's multiple comparisons test. **l, m** Transepithelial electrical resistance (TEER) (**l**) and fluorescence intensity of FITC-dextran (**m**) in the lower chambers of trans-well inserts (0.4 µm) seeded with WT and PEAK1 Y724F Caco-2 cells that express GFP-tagged Src or GFP-Control proteins. Data are presented as mean ± SD, *n* = 3 biological replicates per group. One-way ANOVA, followed by Tukey's multiple comparisons test. The quantified relative expression levels shown in western blots are indicated in red, and represents the results of at least three repeated experiments.

compared to WT controls (Fig. 7h–k). Additionally, following DSS administration, PEAK1, ZO1, and Occludin protein levels decreased in the colon of both WT and *Peak1⁻ᐟ⁻* mice, with ZO-1 and Occludin levels declining more sharply in *Peak1⁻ᐟ⁻* mice (Fig. 7l). DSS treatment also led to a notable increase in Src kinase activity in WT mice; this was less pronounced in *Peak1⁻ᐟ⁻* mice (Fig. 7l). Consistent with our in vitro findings, PEAK1 deletion in mice augmented the interaction between CSK and Src and the binding of ZO-1 to LC-3B in DSS-treated colons (Fig. 7m, n). These findings indicate that the absence of PEAK1 increases susceptibility to DSS-induced colitis by promoting the degradation of ZO-1 in vivo.

## Discussion

Tight junctions, essential intercellular protein complexes in epithelial cells, regulate solute and water passage, ensuring tissue integrity and selective permeability to maintain overall physiological balance. In this study, we delved into the remodeling function of PEAK1 at intestinal epithelium tight junctions. Functionally, we demonstrated that the absence of PEAK1 disrupts tight junction strands in intestinal epithelial cells, augmenting intestinal permeability and compromising resistance to DSS-induced colitis. Mechanistically, we showed that PEAK1 localizes at tight junctions by anchoring to ZO-1 at its LIR motif, an interaction that is promoted by Src-mediated phosphorylation of Y724 on PEAK1, which in turn impedes autophagy-mediated ZO-1 degradation (Fig. 8). Our findings establish PEAK1 as a tight junction protein and reveal a regulatory mechanism by which PEAK1 maintains the integrity of intestinal tight junctions by preventing ZO-1 autophagy-dependent degradation.

PEAK1 was initially identified in isolated pseudopodia through the enrichment of phosphorylated tyrosine (pY) proteins and was predicted to bind F-actin and localize at focal adhesions[29]. Previous studies have linked the cytoskeleton to tight junctions, particularly via Src, ZO-1, and myosin[42]. While our preliminary data in PEAK1-deficient Caco-2 cells showed no defects in myosin pathways (Supplementary Fig. 9), whether additional effects of PEAK1 in the cytoskeleton could also participate in the regulation of tight junction permeability and ZO-1 stability in the colon needs to be further explored.

PEAK1 is highly expressed in malignant tumors[23,24,26], where it promotes cancer cell migration by reorganizing F-actin[29] and regulating focal adhesion dynamic[28]. Most research has focused on the functions of PEAK1 in pathological conditions, particularly cancer, with minimal attention given to its role in diseases other than cancer. In this study, we provide conclusive experimental evidence that the absence of PEAK1 in intestinal epithelial cells disrupts the architecture of tight junctions and augments susceptibility to experimental colitis (Figs. 1, 7), expanding PEAK1's roles beyond promoting tumor cell proliferation and migration to essential physiological processes.

ZO-1, the first discovered tight junction protein[43], employs its multiple subdomains to recruit other tight junction proteins and

associated proteins to stabilize tight junctions[43,44]. The specific knockout of the ZO-1 gene in intestinal epithelial cells results in a mild increase in macromolecular permeability; however, simultaneous knockout of both ZO-1 and ZO-2 in cultured intestinal epithelial-derived cell lines significantly impairs barrier integrity and claudin recruitment at tight junctions[10,12], indicating functional redundancy between ZO-1 and ZO-2 in maintaining tight junction function. Similar to ZO-1 knockout mice[12], we did not detect significant disruption of tight junction structure or spontaneous intestinal inflammation in *Peak1⁻ᐟ⁻* mice, but we did observe increased intestinal barrier tight junction-dependent permeability, mild impairment of tight junction structure, and impaired intestinal epithelial recovery following DSS-induced damage. In *Peak1⁻ᐟ⁻* mouse intestinal tissues and PEAK1 knockout Caco-2 cells, we observed a reduction in ZO-1 levels, along with a similar decrease in Occludin and Claudin 2. Occludin and Claudin 2, also key junction proteins, interact with ZO-1 to maintain the barrier function of tight junction[45,46]. Its reduction in *Peak1⁻ᐟ⁻* mice may result from the decrease of ZO-1 and mild impairment of the tight junction. Still, additional research is needed to explore the underlying reasons for Occludin and Claudin2 decrease and the enhanced interaction between ZO-1 and Occludin following PEAK1 deletion.

Post-translational modifications critically regulate protein structure, function, localization, and interactions within cells, knowingly shaping cellular processes[15]. Src, a ubiquitously expressed tyrosine kinase, is recognized as a key contributor to intracellular tyrosine phosphorylation. Several tyrosine residues on PEAK1 are predicted to undergo phosphorylation based on bioinformatic analysis and mass spectrometry[47], and previous studies identified PEAK1 as a substrate for Src[29,31]. PEAK1 is a highly tyrosine-phosphorylated protein, with Y635 and Y665 being activated by the SFK family kinases Lyn and Src, respectively[30,48]. We have now confirmed Y724 as a second PEAK1 tyrosine that undergoes Src phosphorylation to promote tight junction integrity.

Previous studies have shown that the interaction between Pragmin (PEAK1 homologue) and CSK establishes a feed-forward loop leading to CSK activation[49]. In our studies, deletion of CSK or overexpression of activated Src does not rescue the phenotype of PEAK1 Y724F mutant in epithelial cells (Figs. 4, 5), indicating that PEAK1 functions downstream of CSK and Src. Interestingly, we found that PEAK1 binds to CSK to positively regulate Src activity, and in turn, promote phosphorylation at its own Y724 site. This uncovered interplay between CSK, Src, and PEAK1, plays a pivotal role in sustaining the stability of tight junctions. However, since both PEAK1 and Pragmin interact with CSK and Src[25,28,50], further experiments are needed to explore the regulatory relationship among these proteins.

Previous studies have independently linked Src, ZO-1 phosphorylation, and autophagy to tight junction regulation[51–53]. However, the precise molecular mechanisms connecting these factors, as well as those governing ZO-1 stability in tight junction integrity, remain

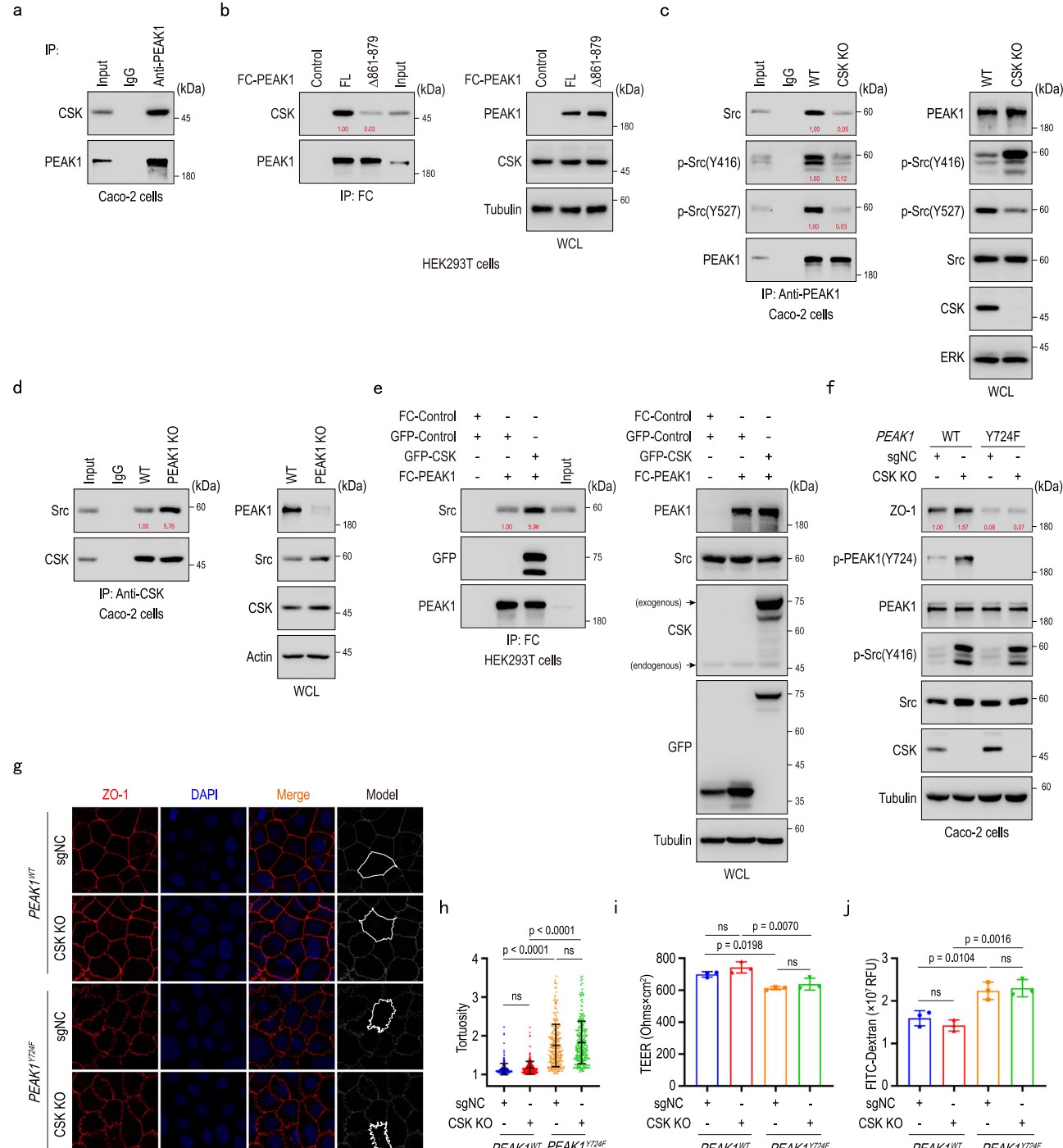

**Fig. 5 | PEAK1-CSK binding regulates Src activity in the maintenance of tight junction integrity. a** Co-immunoprecipitation of Caco-2 cell lysates with anti-PEAK1 antibodies to detect CSK binding to PEAK1. **b** Co-immunoprecipitation analysis for PEAK1 and CSK binding using protein lysates from HEK293T cells overexpressing either FC-tagged PEAK1 (FL) or FC-tagged PEAK1 with an 861-879 amino acid deletion (Δ 861-879). **c** Co-immunoprecipitation of wild-type (WT) and *CSK* knockout (CSK KO) Caco-2 cell lysates, showing reduced PEAK1-Src interaction in the absence of CSK. **d** Co-immunoprecipitation analysis using anti-CSK antibodies to detect the interaction between CSK and Src in WT and PEAK1 knockout Caco-2 cells. **e** Co-immunoprecipitation of HEK293T cell lysates co-overexpressing FC-tagged PEAK1 and GFP-CSK, showing that CSK overexpression promotes PEAK1-Src binding. **f** Western blot analysis of WT and PEAK1 Y724F mutant Caco-2 cells expressing CSK (sgNC controls) or lacking CSK (CSK KO). **g, h** Representative

confocal images (**g**) and barrier tortuosity quantification (**h**) of WT and PEAK1 Y724F Caco-2 cells expressing CSK (sgNC, WT controls) or lacking CSK (CSK KO). Scale bars, 10 μm. Data are shown as mean ± SD. PEAK1^WT + sgNC, *n* = 310 cells; PEAK1^WT + CSK KO, *n* = 326 cells; PEAK1^Y724F + sgNC, *n* = 257 cells; PEAK1^Y724F + sgNC, *n* = 257 cells. Kruskal-Wallis test, followed by Dunn's multiple comparisons test. **i, j** Transepithelial electrical resistance (TEER) (**i**) and fluorescence intensity of FITC-dextran (**j**) in the lower chambers of trans-well inserts (0.4 μm) seeded with WT and PEAK1 Y724F Caco-2 cells, expressing CSK (sgNC controls) or lacking CSK (CSK KO). Data are presented as mean ± SD for three biological replicates. One-way ANOVA, followed by Tukey's multiple comparisons test. The quantified relative expression levels shown in western blots are indicated in red, and represents the results of at least three repeated experiments.

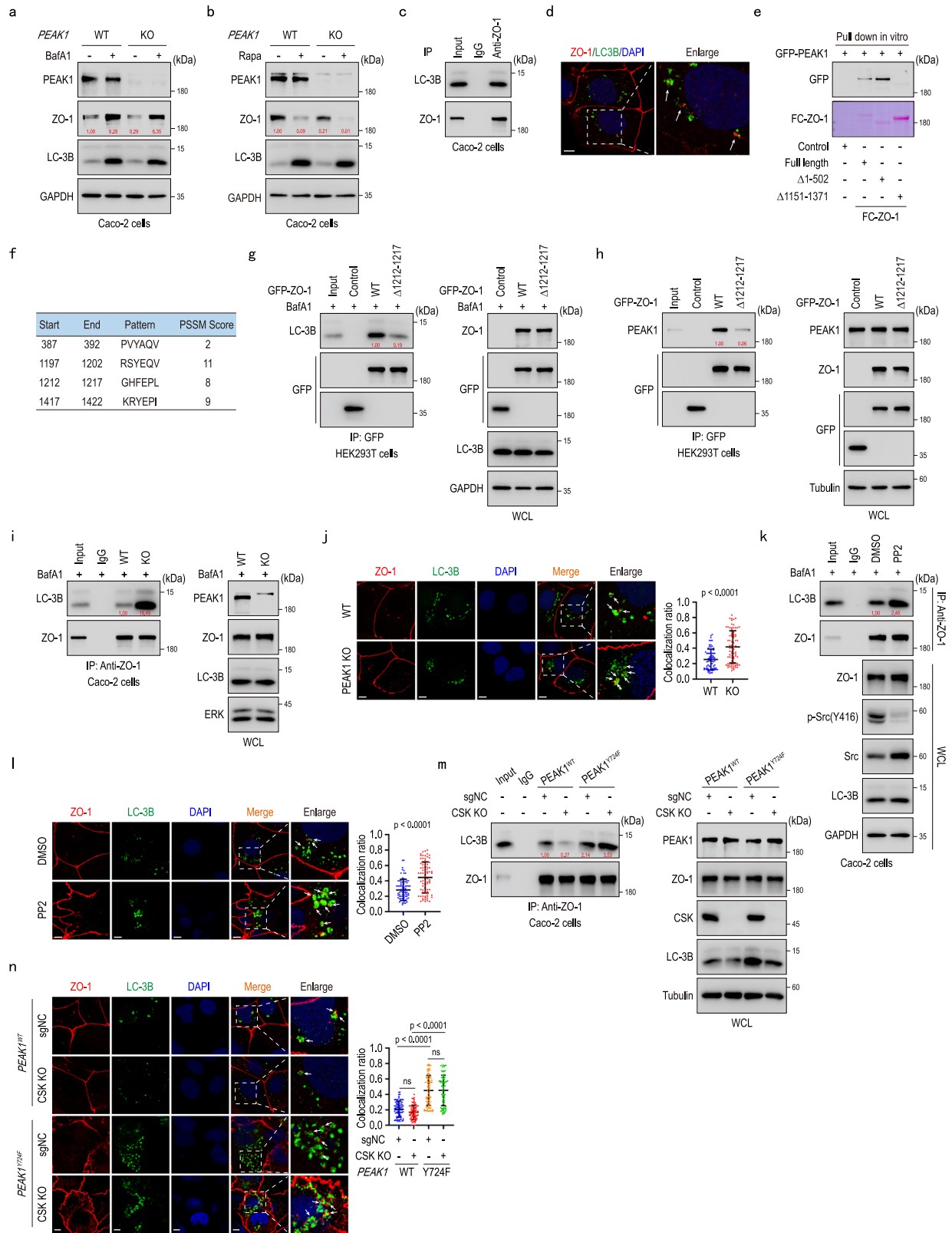

unresolved. In conclusion, our research illuminates the complex regulatory interplay involving PEAK1-CSK-Src/ZO-1 in tight junction dynamics, elucidating an autophagy-mediated degradation mechanism of ZO-1. Our study establishes a foundation for further investigation into the signaling network that governs the interaction between tyrosine kinase and tight junction systems within the intestinal epithelium. This understanding carries substantial implications for

unraveling both the physiological and pathological dimensions of the human gut.

## Methods

### Plasmids

PEAK1 constructs were generated by cloning human *PEAK1* cDNA into the pTriEx-FC, pTriEx-GFP, and pTriEx-3×Flag vectors, respectively.

**Fig. 6 | PEAK1 deletion triggers ZO-1 degradation by exposing an LC3-interacting region on ZO-1.** **a–b** Representative western blot analysis of wild-type (WT) and *PEAK1* knockout (KO) Caco-2 cells treated with 100 nM Bafilomycin A1 (BafA1, autophagy inhibitor) (a) or 100 μM Rapamycin (Rapa, autophagy activator) (b) for 12 h. **c** Co-immunoprecipitation of Caco-2 cell lysates using anti-ZO-1 antibodies, showing LC-3B binding to ZO-1. **d** Immunofluorescence images showing colocalization of ZO-1 and LC-3B in autophagosomes of Caco-2 cells treated with 100 nM BafA1. Scale bars, 10 μm. **e** In vitro pull-down assays using different FC-tagged ZO-1 mutants and GFP-tagged PEAK1. FC-tagged ZO-1 deletions are visualized by Ponceau staining. **f** Predicted interaction motifs between ZO-1 and LC-3B, identified using an online tool (https://ilir.warwick.ac.uk/). PSSM: Position Specific Scoring Matrix (PSSM). High PSSM scores are assigned to the most frequent residues. **g, h** Co-immunoprecipitation analysis of HEK293T cells transfected with GFP-tagged ZO-1 (WT) or GFP-tagged mutant ZO-1 with a 1212-1217 amino acid deletion (Δ1212-1217), confirming this amino acid region regulates ZO-1 binding to LC-3B (g) and PEAK1 (h). **(i)** Co-immunoprecipitation using an anti-ZO-1 antibody in wild-type (WT) and *PEAK1* knockout (KO) Caco-2 cells treated with 100 nM BafA1 for 12 h. **j** Representative Immunofluorescence images (left panel) and quantification of the ratio of ZO-1 and LC-3B co-localization to total ZO-1 puncta per cell (right panel) in WT and PEAK1 KO Caco-2 cells treated with 100 nM BafA1. Scale bars, 10 μm.

KO = PEAK1 KO. Data are presented as mean ± SD. 79 cells per group were analyzed. Unpaired two-tailed Student's t-test. **k** Co-immunoprecipitation using anti-ZO1 antibodies in Caco-2 cells treated with 100 nM BafA1 for 12 h, followed by an additional treatment with 10 uM PP2 inhibitor or vehicle (DMSO) for 2 h. **(l)** Left panel: immunofluorescence microscopy images detecting ZO-1 (red) and LC-3B (green) in Caco-2 cells treated with 100 nM BafA1 for 12 h, followed by treatment with 10 uM PP2 inhibitor or vehicle (DMSO) for 2 h. Scale bars, 10 μm. Right panel: quantification of the ratios of ZO-1 and LC-3B co-localization to total ZO-1 puncta per cell. Data are presented as mean ± SD. WT, *n* = 79 cells per group. Unpaired two-tailed Student's t-test. **m** Co-immunoprecipitation analysis in the protein lysates from WT and Y724F mutant Caco-2 cells expressing CSK (sgNC controls) or lacking CSK (CSK KO) and treated with 100 nM BafA1 for 12 h. **n** Representative confocal images showing ZO-1 (red) and LC-3B (green) in WT and Y724F mutant Caco-2 cells expressing CSK (sgNC controls) or lacking CSK (CSK KO) and treated with 100 nM BafA1 for 12 h. Scale bars, 10 μm. Right panel: quantification of the ratios of ZO-1 and LC-3B co-localization to total ZO-1 puncta per cell. Data are presented as mean ± SD. *n* = 79 cells per group. Kruskal-Wallis test, followed by Dunn's multiple comparisons test. The quantified relative expression levels shown in western blots are indicated in red, and represents the results of at least three repeated experiments.

Mouse PEAK1 was generated by inserting mouse *Peak1* cDNA into the pTriEx-GFP vector. The lentiviral vector expressing PEAK1 was generated by cloning human PEAK1 cDNA into CMV-GFP vectors. To examine specific regions of PEAK1, truncations 1-355, 334-800, 785-1271, and 1260-1746 were generated by incorporating the corresponding human cDNA into pTriEx-FC using FC-PEAK1 as the template. Similarly, ZO-1 full-length and truncations 1-502, 503-876, 877-1150, 1151-1390, 1391-1748 were constructed by introducing human ZO-1 cDNA into the pTriEx-FC and pTriEx-GFP vectors. Additional constructs, such as GFP-Occludin, GFP-Src, FC-Src, and GFP-CSK, were generated by inserting human Occludin, Src, and CSK cDNA into the CMV-GFP or pTriex-Fc vector. GST-Src was generated by inserting human Src into the pET-28a vector.

Deletion and point mutant constructs were created using overlap extension PCR and site-directed mutagenesis. Specifically, PEAK1 mutants S718/719 A, S723A, S725/726/727 A, S729/730 A, 8×S-A, 714-731 deletion (Δ 714-731), 717-722 deletion (Δ 717-722), 720-726 deletion (Δ 720-726), 725-731 deletion (Δ 725-731), 861-879 deletion (Δ 861-879) were generated using FC-PEAK1 as the template. The PEAK1 mutant Y724F was created using GFP-PEAK1 or FC-PEAK1 as templates. ZO-1 mutants, including 1-502 deletion (Δ 387-392) and 1151-1371 deletion (Δ 1151-1371), were generated using FC-ZO-1 as the template. 387-392 deletion (Δ 387-392), 1197-1202 deletion (Δ 1197-1202), 1212-1217 (Δ 1212-1217), and 1417-1422 deletion (Δ 1417-1422) were generated using GFP-ZO-1 as templates. The Src mutants K298M and Y530F were created using GFP-Src as the template. Primer details are provided in Supplementary Table 1.

For experimental procedures, pMD2.0 G (#12259), psPAX2 (#12260), and lentiCRISPR-v2 (#52961) were purchased from Addgene. pL-CRISPR.EFS.GFP vector (Addgene #57818) was a gift from Zhengzhou Ying at Chongqing Medical University.

## Antibodies and other reagents

Anti-PEAK1 antibodies were purchased from Novus Biologicals (WB 1:1,000, IHC: 1:100, NBP1-91052) and Cell Signaling Technology (1:1,000, #72908). Anti-ZO-1 antibodies were purchased from Invitrogen (WB 1:1,000, IF 1:100, 33-9100) and Proteintech (WB 1:2,000, 21773-1-AP). Antibodies against the following proteins were purchased from Invitrogen: ZO-1 Alexa Fluor™ 555 (IF 1:100, MA3-39100-A555), Claudin-2 (WB 1:1,000, IF 1:100, 32-5600), Goat anti-Mouse IgG (H + L) Alexa Fluor® 488 Secondary Antibody (1:500, A-11029), Goat anti-Rabbit IgG (H + L) Alexa Fluor® 488 Secondary Antibody (1:500, A-11034), Goat anti-Mouse IgG (H + L) Alexa Fluor® 555 Secondary Antibody (1:500, A-21424), Goat anti-Rabbit IgG (H + L) Alexa Fluor® 555

Secondary Antibody (1:500, A-21429). Antibodies against the following proteins were purchased from Cell Signaling Technology: Src (WB 1:1,000, #2108), Phospho-Src (Tyr416) (1:1,000, #2101), Phospho-Src (Tyr527) (WB 1:1,000, #2105), ERK1/2 (WB 1:1,000, #4695), LC-3B (WB 1:1,000, #2775), Phospho-Myosin Light Chain 2 (Ser19) (WB 1:1,000, #3671), Myosin Light Chain 2 (WB 1:1,000, #8505), Conformation Specific Mouse Anti-Rabbit IgG (WB 1:2,000, #5127S). Antibodies against the following proteins were purchased from Bioworld: Occludin (WB 1:1,000, AP0765), GFP (WB 1:2,000, MB9233), and E-cadherin (WB 1:1,000, BS1098). Antibodies against the following proteins were purchased from Abcam: GFP (IF 1:100, ab290), Occludin (WB 1:1,000, ab216327), and FC (WB 1:2,000, ab190492). Antibodies against the following proteins were purchased from Proteintech: CSK (WB 1:1,000, 17720-1-AP), Claudin 4 (WB 1:2,000, IF 1:200, 16195-1-AP), and F4/80 (IF 1:200, 28463-1-AP). Antibodies against LC-3 (IF 1:100, 4E12) for immunofluorescence were purchased from MBL Life Science. Anti-phosphotyrosine (4G10) was purchased from Millipore (WB 1:2,000, 05-321). Antibodies against β-actin (WB 1:2,000, HC201-01) and β-tubulin (WB 1:2,000, HC101-01) were purchased from Transgen. Antibodies against Flag (WB 1:4,000, AE005) and GAPDH (WB 1:4,000, AC002) were purchased from Abclonal. Goat Anti-Mouse IgG, HRP (1:5,000, CW0102) and Goat Anti-Rabbit IgG, HRP (1:5,000, CW0103) were purchased from CWBIO. The anti-GFP magnetic bead (L-1016) was purchased from Bio-Linkedin. Protein A Sepharose (17-0780-01) was purchased from GE Healthcare. Dovitinib lactate (HY-10207), STI571 Mesylate (HY-50946), Momelotinib (HY-10961), MG-132 (HY-13259), Bafilomycin A1 (HY-100558), and Rapamycin (HY-10219) were purchased from Med. Chem. Express. PP2 inhibitor (P0042) was purchased from Sigma-Aldrich.

## Phospho-PEAK1 Y724 antibody generation

To generate a phospho-specific antibody targeting the Y724 phosphorylation site of PEAK1, a synthetic phosphopeptide with the sequence CLNRGQSSPQRSY(p)SSSH was meticulously designed, incorporating flanking amino acids to enhance epitope specificity. This phosphopeptide was conjugated to keyhole limpet hemocyanin (KLH) using glutaraldehyde as a crosslinker to boost immunogenicity. Rabbits were immunized with the KLH-phosphopeptide conjugate, with booster injections administered every two weeks to strengthen the immune response. Throughout the immunization process, serum samples were collected and analyzed using enzyme-linked immunosorbent assay (ELISA) to monitor and confirm the generation of a robust antibody response specific to the phospho-Y724 site. Following the completion of the immunization schedule, antibodies were

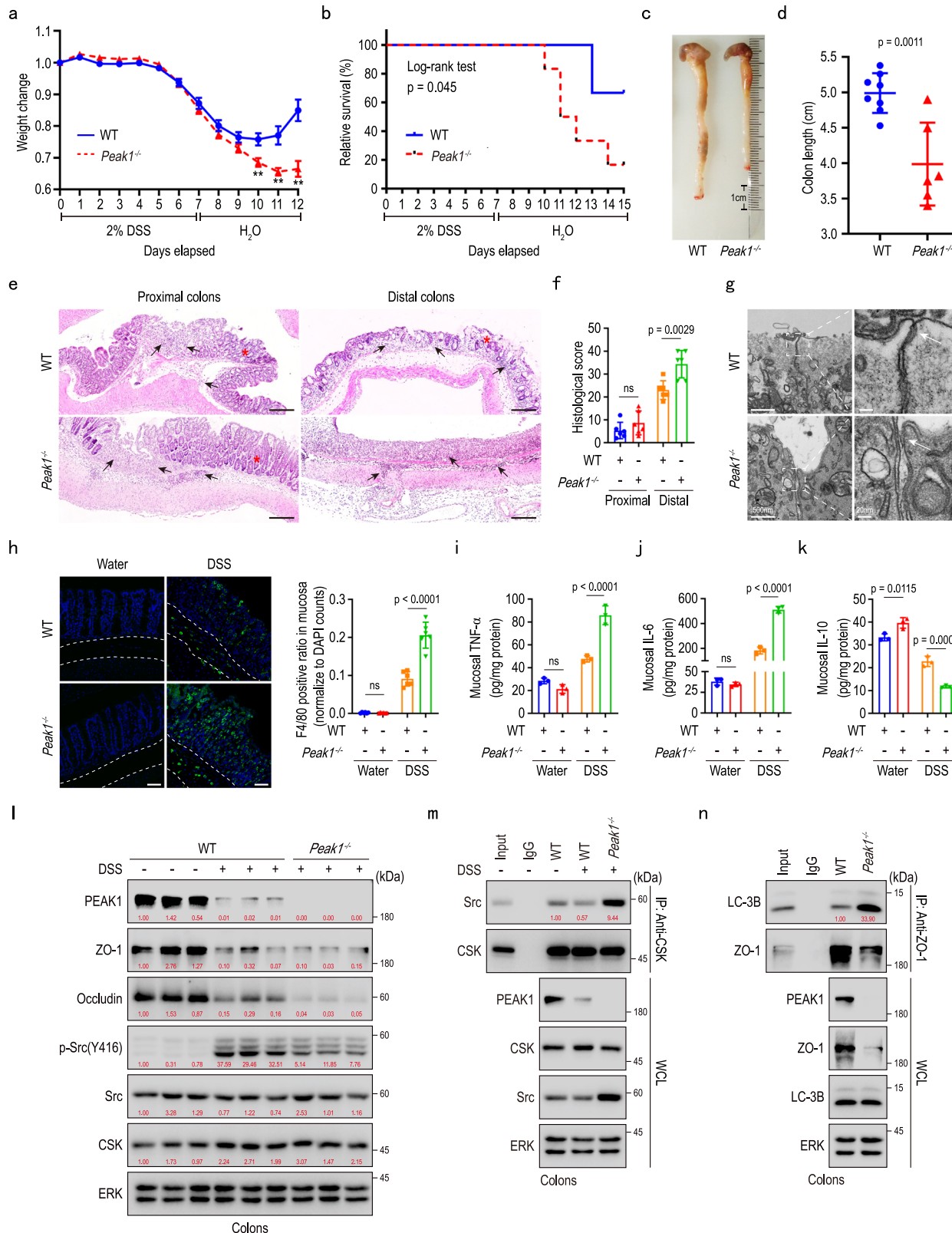

harvested and subjected to purification through affinity chromatography using a phosphopeptide column, followed by characterization for specificity via Western blot.

## Mice

All mouse studies were approved by the Institutional Animal Care and Use Committee (IACUC) of Chongqing Medical University. *Peak1⁻/⁻*

C57BL/6 J mice were designed and constructed by the Biocytogen company and subsequently re-derived at the animal center of Chongqing Medical University (Chongqing, China). *Peak1* homozygous knockout (*Peak1⁻/⁻*) mice were generated by breeding heterozygous (*Peak1⁺/⁻*) mice, and their wild-type (*Peak1⁺/⁺*) littermates were used as controls. *Peak1⁻/⁻* mice were born at the expected Mendelian ratios and displayed no overt phenotypes. The mice were housed in

**Fig. 7 | PEAK1 deletion increases susceptibility to DSS-induced colitis.**
**a** Changes in body weight over time for both WT and *Peak1⁻/⁻* mice, expressed as a percentage of their initial body weight at day 0. Data are mean ± SEM. *n* = 12 mice per group. Unpaired two-tailed Student's t-test. ** represent *p* < 0.01.
**b** Kaplan–Meier survival curve for wild-type (WT) and *Peak1⁻/⁻* mice treated with 2% DSS. WT, *n* = 7 mice; *Peak1⁻/⁻*, *n* = 6 mice. **c, d** Representative photographs of colons (**c**) and quantification of colon length (**d**) in WT and *Peak1⁻/⁻* mice on day 12 after initiating 2% DSS administration. Data are presented as mean ± SD. WT, *n* = 8 mice; *Peak1⁻/⁻*, *n* = 6 mice. Unpaired two-tailed Student's t-test. **e** Representative H&E-stained images of colon cross-sections from WT and *Peak1⁻/⁻* mice on day 12 after 2% DSS administration. Scale bars, 200 μm. Black arrows denote immune cell infiltration, and red asterisks indicate intestinal crypts. **f** Colitis scores from histological analysis of colon tissue, as shown in e. Data are presented as mean ± SD, *n* = 6 mice per group. One-way ANOVA, followed by Tukey's multiple comparisons test. **g** TEM images showing tight junctions in colon cross-sections from WT and *Peak1⁻/⁻* mice

on day 12 after 2% DSS administration. Left panel, scale bars, 500 nm. Right panel, scale bars, 20 nm. **h** Representative confocal images (left panel) and quantification (right panel) of F4/80+ macrophages in the epithelium of the colon tissues from WT and *Peak1⁻/⁻* mice treated with water or on day 12 after 2% DSS administration. The area between the white lines indicates the submucosa. Scale bars, 50 μm. Data are presented as mean ± SD. *n* = 6 mice per group. One-way ANOVA, followed by Tukey's multiple comparisons test. **i–k** Levels of TNFα (**i**), IL-6 (**j**), and IL-10 (**k**) in colon tissues from untreated and DSS-treated WT and *Peak1⁻/⁻* mice on day 12, as determined by ELISA. Data are shown as mean ± SD for three biological replicates. One-way ANOVA, followed by Tukey's multiple comparisons test. **l–n** Western blot (**l**) and immunoprecipitation (**m, n**) analyses using the indicated antibodies, in tissue lysates from colons collected on day 12 from wild-type (WT) or *Peak1⁻/⁻* mice treated with or without 2% DSS. The quantified relative expression levels shown in western blots are indicated in red, and represents the results of at least three repeated experiments.

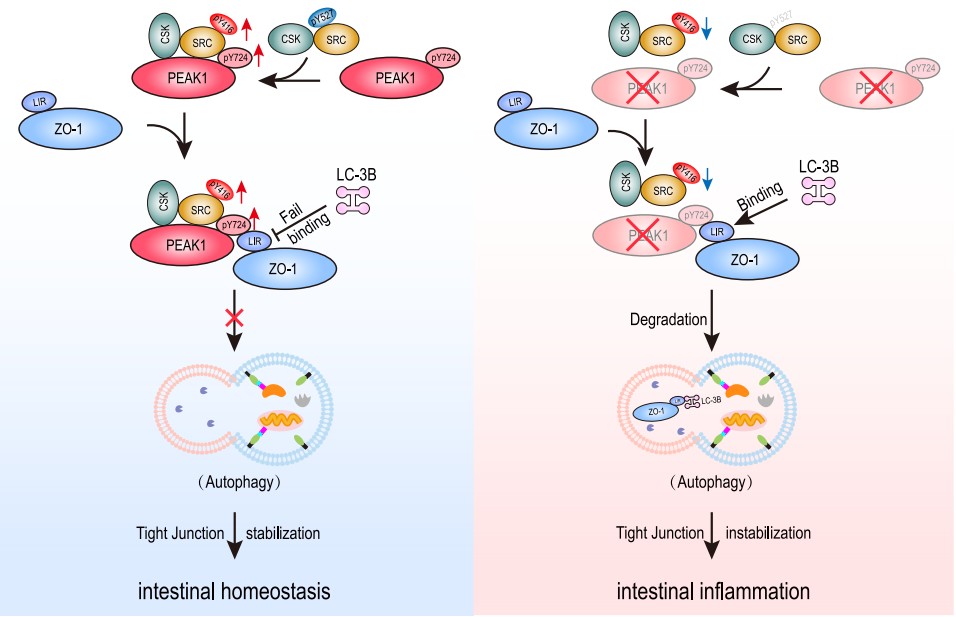

**Fig. 8 | Schematic illustration of the proposed molecular mechanism.** Under physiological conditions, the PEAK1/CSK/Src/ZO-1 complex at cell-cell contact sites maintains the integrity of tight junctions. Specifically, PEAK1 interacts with CSK to sustain Src activity, while activated Src phosphorylates PEAK1 at tyrosine Y724, facilitating PEAK1's interaction with ZO-1. This interaction masks the LC-3 binding LIR motif on ZO-1, thereby preserving ZO-1 expression at the tight junctions by preventing ZO-1/LC-3B binding and subsequent ZO-1 autophagy-mediated degradation. The graphic was created by us using Adobe Illustrator.

specific pathogen-free conditions at the animal center of Chongqing Medical University and were subject to a strict 12-hour light-dark cycle (lights on at 8 am and off at 8 pm). They were accommodated in cages with a maximum occupancy of five animals. In all experiments, co-caged littermates were used, and analysis was performed using 8-12 weeks old male mice.

## Mouse genotyping
Genomic DNA was extracted from mouse tails by boiling the tail biopsies at 95 °C for 4 h with buffer I (25 mM NaOH, 0.2 mM EDTA), followed by the addition of buffer II (4 mM Tris-HCl, pH 5.5). After centrifugation at 12,000 × g for 10 min, the supernatants were collected for PCR genotyping using Dream Taq Green PCR Master Mix (Thermo Fisher, K1082). Amplicons were analyzed by agarose gel electrophoresis. Primer details are provided in Supplementary Table 2.

## Cell culture, treatments, and transfections
HEK293T, Caco-2, and NCM460 cells were cultured in Dulbecco's modified Eagle's medium (DMEM) (Thermo Fisher, C11995500BT),

supplemented with 10% fetal bovine serum (Bio-Channel, BC-SE-FBS01) and 100 units/ml penicillin-streptomycin (Cytiva, SV30010). For Caco-2 cells used in immunofluorescence studies, the DMEM medium contained 1.0 g/L D-(+)-glucose (Thermo Fisher, C11885500BT), 10% fetal bovine serum, and 100 units/ml penicillin-streptomycin[7]. Cultures were maintained in a humidified chamber at 37 °C with 5% CO2. HEK293T, Caco-2, and NCM460 cells were obtained from ATCC and underwent transfection using Lipofectamine 2000 (Invitrogen, 11668019) or TurboFect Transfection Reagent (Thermo Fisher, R0531) following the manufacturer's instructions.

## Lentiviral vectors and stable cell lines
For lentiviral production, HEK293T cells were co-transfected with each lentiviral plasmid along with the corresponding helper plasmids. The virus-containing medium was harvested 48 h post-transfection, subjected to centrifugation at 1200 × g for 5 min, and then filtered through 0.45 μm filters (Millipore, SLHV033RB). Subsequently, cells were transduced with lentiviral particles in the presence of 8 μg/ml polybrene (Sigma-Aldrich, H9268) for 20–24 h and underwent selection with 100 μg/ml puromycin (Invivogen, ant-pr-1) for one week.

## Transient RNAi knockdown

For transient knockdown, cells were transfected with siRNAs (Gene-pharma) using Lipofectamine 2000 (Invitrogen Life Technologies, 11668019) according to the manufacturer's instructions. The siRNA sequences were as follows: negative control, sense 5′-UUCUCCGAAC-GUGUCACGUTT-3′, antisense 5′-ACGUGACACGUUCGGAGAATT-3′. *PEAK1*, sense 5′-UUAUUAAGUGGAU CUUUUGCATT-3′, antisense 5′-UGCAAAAGAUCCACUUAAUAATT-3′.

## CRISPR–Cas9-mediated genome editing

Genomic mutations were introduced into cells using the CRISPR–Cas9 system according to standard procedures[54]. Briefly, single-guide RNAs (sgRNAs) were designed to target the genomic area adjacent to mutation sites in PEAK1 (Y724F) using the CRISPR design tool (http:// benchling.com/). The annealed guide RNA oligonucleo-tides were inserted into pL-CRISPR.EFS.GFP vector (Addgene#57818) digested with the BsmBI restriction enzyme. Cells were seeded at 60% confluence, followed by co-transfection of sgRNAs (0.5 μg) and single-stranded oligonucleotide (ssODN, 10 pmol) as a template to introduce mutations. sgRNA targeting sequence for PEAK1 (Y724F): 5′- ctggaac-tatagcttctctg -3′. Single-stranded oligonucleotide (ssODN) sequence for PEAK1 (Y724F):5′-ttttcaaacagcacagagcataaaaggggctcagtggctca-gaaggttcaag agtttaacaactgtctcaacagaggtcagtcttcaccacaAagGagTtTtag tAGcagccacagctccccagcaaagatccagagagccactcaagagcctgtggccaaata-gaaggcactcaggagtctcagatggtgggcagcag-3′. The upper-case letters in the ssODN sequences indicate the mutated nucleotides that will replace the endogenous nucleotides in the genomic DNA of parental cells using the CRISPR–Cas9 system. Twenty-four hours after transfection, cells were trypsinized, diluted into single cells, and seeded into 96-well plates. Genomic DNA was extracted from GFP-positive cells, followed by sequencing of the PCR products spanning the mutation sites. Genotyping was performed by sequencing PCR products amplified from the following primers: PEAK1 forward: 5′-tgtca-caaatcagcacctac-3′; PEAK1 reverse: 5′-attcagcttctgttgacctg-3′.

To generate PEAK1-knockout and CSK-knockout Caco-2 cell lines, single guide RNA (sgRNA) targeting PEAK1 and CSK were cloned by annealing two DNA oligos and ligating into lentiCRISPR v2. Caco-2 cells were infected with lentiCRISPR v2-sgRNA vectors, and after selection in the presence of 100 μg/ml of puromycin for a week, cells were trypsinized, diluted for single cells, and seeded into 96-well plates. The knockout efficiency was measured by western blotting. PEAK1 sgRNA #1: 5′-TGCCCGTGTTCCTGATGCGG-3′[31]; PEAK1 sgRNA #2: 5′- GAAGT-GATTAG TAATGAAGG- 3′ (addgene: BRDN0001147244). CSK sgRNA: 5′- ATTGCCAAGTACAACTTC CA-3′; sgNC: 5′-TTCTCCGAACGTGT-CACGTT-3′.

## Western blotting

Cells from culture dishes were rinsed with cold 1×PBS and extracted with a lysis buffer (50 mM Tris-HCl, pH 7.5, 150 mM NaCl, 1 mM EDTA, 1% (w/v) SDS, 2 mM sodium orthovanadate, 1 mM PMSF, Protease Inhibitor Cocktail (APExBIO, K1007), Phosphatase Inhibitor Cocktail (APExBIO, K1015)). After being boiled for 10 min, lysates were cen-trifuged at 12,000 × $g$ for 10 min, and the supernatant was collected for protein quantification using the Pierce BCA protein assay (Thermo Fisher, 23227). Samples were prepared using 5×loading buffer (200 mM Tris-HCl, pH 6.8, 8% SDS, 40% glycerol, 0.4% bromophenol blue, 5% β-mercaptoethanol) and boiled for 10 min. Samples were electrophoresed in SDS-PAGE gels and transferred onto PVDF (Milli-pore, IPVH00010) using wet transfer at 300 mA for the appropriate time. 5% nonfat dry milk (Cell Signaling Technology, #9999) in TBST was used to block the membrane for 2 h. Blots were incubated over-night at 4 °C with primary antibodies prepared in 3% milk. After three washes with TBST, the membrane was incubated with HRP-conjugated secondary antibody for 1 h at room temperature. Following secondary incubation, the blot was washed three times in TBST. Membranes were incubated with the Western Blot Smart-ECL reagents (Smart-Life-sciences, 30500 or H31500) according to the manufacturer's instruc-tions. Depending on the experiment, GAPDH, β-Tubulin, β-actin, or ERK1/2 was used as the loading control.

## Co-immunoprecipitation

Cells were rinsed with 1×PBS and lysed using RM buffer (50 mM Tris, pH 7.5, 150 mM NaCl, 1 mM EDTA, 0.1% deoxycholate, 1% NP-40, 1 mM Na orthovanadate, 10 mM β-Glycerolphosphate, 10 mM sodium fluor-ide, 1 mM PMSF, Protease Inhibitor Cocktail, Phosphatase Inhibitor Cocktail). After ultrasonic treatment, cell lysates were centrifuged at 12,000 × $g$ at 4 °C for 15 min. For immunoprecipitation using anti-bodies, 1 mg of cell protein lysate was incubated with antibodies for 4 h at 4 °C on a rotator. Lysate/antibody mixtures were incubated with 200 μL protein A sepharose (GE Healthcare, 17-0780-01) for 4 h at 4 °C on a rotator and subsequently washed three times using 500 μL RM buffer. Proteins were eluted at 95 °C for 10 min with 2×loading buffer. For immunoprecipitation using anti-GFP magnetic beads, cell lysates were incubated with 5 μl of beads and incubated overnight at 4 °C on a rotator. Beads were washed three times in RM buffer and eluted as described above. Eluates were analyzed by Western blotting as described above.

## Sample preparation for LC-MS/MS analysis

Immunoprecipitated elutes, obtained using our co-immunoprecipitation protocol, were separated by SDS-PAGE and stained with Coomassie Brilliant Blue to visualize specific bands. The bands of interest, excluding the antibody heavy chain, were excised into small pieces (approximately 1 to 2 mm³) for in-gel tryptic diges-tion. To prepare for digestion, the excised gel pieces, corresponding to the target protein's molecular mass, were first destained with 50% acetonitrile in 50 mM ammonium bicarbonate (NH₄HCO₃), then dehydrated with 100% acetonitrile for 5 min. The gel pieces were then incubated with 10 mM TCEP at 37 °C for 30 min to reduce disulfide bonds. Afterward, the gel was dehydrated again with 100% acetonitrile and incubated with 25 mM iodoacetamide at room temperature for 30 min in the dark to alkylate cysteine residues. Following this, the gel pieces were washed with 50 mM NH₄HCO₃, dehydrated again with 100% acetonitrile, and then rehydrated before being digested over-night with 2 μg trypsin in 50 mM NH₄HCO₃ at 37 °C. After digestion, peptides were extracted from the gel with 50% acetonitrile/0.1% formic acid, dried in a SpeedVacuum concentrator, and resuspended in 0.1% formic acid for LC-MS/MS analysis.

## LC-MS/MS for analysis of proteins

The isolated peptides were identified using a Nano-liquid chromato-graphy tandem Q Exactive HF Orbitrap mass spectrometer equipped with an EASY Spray nanospray source (Thermo Scientific). The liquid chromatography system used was the EASY nLC-1200 System (Thermo Scientific). The Nano liquid chromatography gradient was run at a constant flow rate of 400 nl/min, and consisted of the following: an increase from 2% to 7% mobile phase B (0.1% formic acid in 80% acetonitrile) over 1 min, followed by 7% to 35% for 35 min, 35% to 55% for 9 min, a climb to 100% in 7 min, and held at 100% for the last 8 min.

The EASY Spray voltage was set to 2 kV, with the heated capillary temperature set to 275 °C. The scan sequence of the mass spectro-meter was modified based on the original TopTen™ method[55,56]. The analysis included a full scan recorded between 350–1500 Da at a resolution of 60,000, followed by MS/MS scans at a resolution of 15,000 to generate product ion spectra for determining amino acid sequences. The instrument scans focused on the fifteen most abun-dant peaks in the spectrum. The AGC Target ion number was set at 3e6 ions for full scan and 1e5 ions for MS² mode. Maximum ion injection time was set at 50 ms for full scan and 50 ms for MS² mode. Micro scan number was set at 1 for both full scan and MS² scan. The HCD

fragmentation energy (N)CE/stepped NCE was set to 27 and an isolation window of 1.0 m/z. Singly charged ions were excluded from MS[2]. Dynamic exclusion was enabled with a repeat count of 1 within 15 s, excluding isotopes.

Mass spectra were processed and searched using Proteome Discoverer (version 2.4, Thermo Scientific) against the human SwissProt protein database (release 2023-09). Trypsin was specified as a cleavage enzyme, allowing up to 2 missing cleavages. The precursor ion mass tolerance was set to 10 ppm, and the fragment ion mass tolerance was set to 0.01 Da. Carbamidomethylation of cysteine was specified as a fixed modification, while oxidation of methionine, acetylation at the protein N-terminus, and phosphorylation were set as variable modifications. Peptide confidence was set to high.

### Protein expression and purification

pET-28a-GST-Src$^{WT}$ and pET-28a-GST-Src$^{K298M}$ were transformed into chemically competent *Escherichia coli* BL21(DE3) generated in-house. One colony was picked and cultured in LB medium (5% yeast extract, 10% tryptone, 10% NaCl) containing 50 µg/mL kanamycin at 37 °C with shaking at 210 rpm. The bacteria suspension was allowed to reach an OD 600 in the range of 0.6–0.8 before adding IPTG to a final concentration of 0.5 mM. Bacteria were grown overnight at 18°C with shaking at 170 rpm. For obtaining bacterial lysate containing proteins, bacteria pellets were collected and resuspended in cold PBS containing 1 mM PMSF and 1% Triton X-100, then lysed using a high-pressure homogenizer, and centrifuged at 12,000 × g for 20 min at 4 °C. The protein was purified using GST 4FF (Sangon Biotech, C600913) in accordance with the manufacturer's instructions. The purified protein was stored at −80 °C after three times dialysis (Solarbo, MWCO3500) in PBS at 4 °C for 12 h. Purified proteins were verified by SDS-PAGE.

For eukaryotic purified proteins, plasmids encoding GFP-ZO-1, FC-PEAK1, or FC-PEAK1$^{Y724F}$ were transfected into HEK293T cells using TurboFect Transfection reagent following the manufacturer's instructions. Target proteins were harvested 48-72 h post-transfection using RM buffers, followed by immunoprecipitation with protein A sepharose and anti-GFP magnetic beads. Proteins were eluted by incubating the beads with 0.1 M glycine buffer (pH 2.5) for 5 min at room temperature. The eluted fraction was then collected by centrifugation at 1000 × *g* for 1 min. Immediately after, the protein was neutralized with 1 M Tris-HCl (pH 8.0) to adjust the pH to 7.4. The proteins were quantified using the BCA assay, and their quality was assessed by SDS-PAGE and Coomassie Brilliant Blue staining.

### In vitro kinase assay

HEK293T cells were transfected with FC-PEAK1$^{WT}$ and FC-PEAK1$^{Y724F}$, and after 48 h, lysed in RM buffer with 0.1% SDS. Proteins immobilized on Protein A Sepharose, as described above, were washed three times with lysis buffer, twice kinase buffer (25 mM Tris, pH 7.5, 5 mM β-Glycerophosphate, 2 mM DTT, 0.1 mM Na3VO4), and preincubated with kinase buffer containing 2 mM MnCl2, 10 mM MgCl2 and 2 mM ATP at 30 °C for 10 min. Purified GST-Src$^{WT}$ or GST-Src$^{K298M}$ were then added, and the mixture was incubated at 30 °C for 30 min with gentle agitation. The reactions were stopped by the addition of 2× SDS loading buffer and analysed using SDS−PAGE and immunoblotting using antibodies as described before.

### Pull-down assay

For pull-down assays in cell lysates, HEK293T cells were transfected with 3×Flag-PEAK1 or FC-ZO-1 mutants and lysed in RM buffer 48 h post-transfection. After sonication, the lysates were centrifuged at 12,000 × g at 4 °C for 15 min, and the supernatants were collected. Lysates containing 3×Flag-PEAK1 were mixed with lysates containing FC-ZO-1 mutants and incubated at 4 °C overnight. The mixtures were then incubated with 200 µL Protein A Sepharose for 4 h at 4 °C on a

rotator. After washing the lysate-bead mixtures three times with 500 µL RM buffer, proteins were eluted at 95 °C for 10 min with 2× loading buffer. Eluates were analyzed by Western blotting as described above.

For the in vitro pull-down assay, Protein A Sepharose beads were pre-blocked with IP buffer containing 10% FBS at 4 °C overnight with continuous rotation. The blocked beads were then incubated with 10 µg of either FC-tagged PEAK1$^{WT}$ or FC-tagged PEAK1$^{Y724F}$ at 4 °C for 3 h with rotation. After thorough washing with IP buffer, the beads were incubated with increasing concentrations of GFP-tagged ZO-1 (0, 1, 2, and 4 µg) for 4 h at 4 °C with rotation. After three additional washes with IP buffer to remove non-specifically bound proteins, the samples were analyzed by immunoblotting to detect the bound prey proteins.

### Trans-epithelial electrical resistance (TEER) and FITC-Dextran permeability in vitro

Caco-2 cells were plated at a density of ~2×10$^5$ cells/cm² on 0.4 µm pore transwell inserts (Labselect, 14211) and maintained in complete medium until the establishment of tight junctions. TEER was determined by measuring the resistance across the monolayer using a VOM3 Epithelial Volt/Ohm Meter with STX2-PLUS (WPI). The resistance value, measured in ohms (Ω), was obtained by subtracting the TEER value of the blank insert and multiplying the difference by the growth surface area of the filter. Then, cells were washed with PBS three times, and FITC-Dextran (1 mg/ml, 4 kDa) dissolved in phenol red-free DMEM complete media (10% fetal bovine serum and 100 units/ml penicillin-streptomycin) was added to the top chamber. Phenol red-free DMEM complete media was added to the lower chamber. After 1 h, 100 µL of DMEM was collected from the lower chamber, and FITC levels were determined by SpectraMax® iD5.

### In vivo intestinal permeability studies

Intestinal permeability was assessed using fluorescein isothiocyanate-dextran (FITC-Dextran, average molecular weight 4,000; Sigma-Aldrich, 46944), Rhodamine B isothiocyanate-dextran (average molecular weight ~70,000; Sigma-Aldrich, R9379), and creatinine (molecular weight 113; Sigma-Aldrich, 1.05206), following standard procedures[57]. Briefly, 22 mg/mL of FITC-Dextran, 16 mg/mL of Rhodamine B isothiocyanate-dextran, and 40 mg/mL of creatinine were dissolved in sterile saline. 0.5 mL volume of this solution was administered to the mice via oral gavage using a 20-gauge feeding needle. Five hours later, blood samples were collected from sacrificed mice, and the isolated serum was diluted 1:1 with sterile saline. FITC and Rhodamine B concentrations were measured using a SpectraMax® iD5 microplate reader with excitation/emission wavelengths of 485/528 nm and 543/580 nm, respectively. Creatinine levels were quantified using a cobas® 8000 modular analyzer (HITACHI).

### Isolation of intestinal epithelial cells

Colonic epithelial cells were isolated according to standard procedures[7]. In brief, colons were dissected, washed with PBS, and cut into small pieces. Colon segments were incubated in HBSS supplemented with 5 mM EDTA and 0.5 mM DTT for 30 min at 37 °C with gentle shaking. The epithelial cells in the supernatants were filtered through a 70 µm cell strainer and washed twice. After adding 1% SDS lysis buffer to the isolated epithelial cells, lysates were sonicated until clear and used for western blotting.

### Induction of DSS-induced colitis

2% (w/v) dextran sulfate sodium (DSS, MP Biomedicals, 160110, molecular weight 36,000–50,000 kDa) was dissolved in the drinking water of mice. 8- to 12-week-old *Peak1$^{-/-}$* and WT male mice were exposed to 2.0% DSS for 7 days (fresh DSS solution was provided every day), followed by drinking water for 8 days. The mice were monitored

each day for records of morbidity and body weight. Induction of colitis was determined by weight loss, fecal blood, diarrhea, and, upon autopsy, weight and length of colon. Mice were euthanized if their body weight loss was ≥ 35%.

## Immunohistochemical analyses

Immunohistochemical staining was performed according to the kit (ZSGB-BIO, PV-9001) manufacturer's instructions. In brief, dissected colon tissues were fixed in 4% paraformaldehyde solution and paraffin-embedded (FFPE). Five-micrometer sections were dewaxed in xylene and rehydrated in graded alcohol baths. Microwave antigen retrieval was performed for 15 min in citrate buffer (pH 6.0). The slides were treated with 0.1% Triton X-100, washed with PBS three times, and incubated with hydrogen-peroxide solution for 15 min to block endogenous peroxidase activity. After washing in PBS, the slides were incubated for 1 h with normal sheep serum to eliminate non-specific staining and then incubated with anti-PEAK1 antibody, (1:100, Novus Biologicals, NBP1-91052,) overnight at 4 °C. After washing in PBS containing 0.01% Tween-20 (PBST), slides were incubated with biotin labeled secondary antibody and finally developed using 3,30-diaminobenzidine-tetrahydrochloride (DAB, ZSGB-BIO, ZLI-9017). The nuclei were counter-stained with hematoxylin (BASO, BA-4097). The slides were visualized, and images were acquired using a Leica DM6 B microscope.

## Histopathology of colon

On day 9 after DSS treatment started, the entire colon was dissected and its length measured. Colons were washed, fixed in 4% buffered paraformaldehyde, embedded in paraffin, and micro-sectioned for hematoxylin & eosin (H&E) staining. Histology was scored by a pathologist in a blinded fashion as described[58]. The following independent parameters associated with colitis were assessed in both proximal as well as distal colon regions: severity of inflammation (I), depth of lesions (E), crypt damage (C), and range of lesions (P). Severity was scored on a scale of 0–3, with none = 0, slight = 1, moderate = 2, or severe =3. Depth of lesions was scored on a scale of 0–3, with none = 0, submucosal = 1, muscularis = 2, or serosal = 3. Crypt damage was scored on a scale of 0–4, with no crypt damage = 0, basal one-third damaged = 1, basal two-thirds damaged = 2, only surface epithelium intact = 3, or entire crypt and epithelium lost = 4. The range of lesions was scored on a scale of 1–4, with 1–25% = 1, 26–50% = 2, 51–75% = 3, or 76–100% = 4. The combined histological score = (I + E + C) × P, ranged from 0 (no changes) to 40 (extensive infiltration and tissue damage).

## Immunofluorescence

Culture cells or colon tissues from frozen sections were washed twice with PBS, incubated in 4% (w/v) paraformaldehyde for 15 min, and washed three times with PBS. Fixed cells or tissues were then incubated in 0.1% TritonX-100, washed three times with PBS, and blocked with 10% BSA for 2 h at room temperature. Cells were incubated in appropriate concentrations of primary antibody in 2% BSA overnight at 4 °C. Cells and slides were washed three times in 0.1‰ PBST and incubated with fluorescently-labeled secondary antibodies and DAPI in 2% BSA for 1 h. Cells and slides were washed three times in 0.1‰ PBST and mounted on coverslips containing polyvinylpyrrolidone medium (Beyotime, P0123). Pictures were acquired with a Zeiss LSM710 confocal microscope at objective ×63. Tortuosity was quantified using Image J software.

## qRT-PCR

Total cellular RNA was extracted and purified using the Biospin Total RNA Extraction kit (Hangzhou Bioer Technology, BSC63S1) and reverse transcribed to cDNA using the RevertAid First Strand cDNA Synthesis Kit (Thermo Fisher, K1622) according to the manufacturer's

specifications. Quantitative PCR was performed in Roche Light-Cycler®96 using UltraSYBR Mixture (CWBIO, CW0957M). Samples were run in a 25 μl mixture using the following PCR program: initial denaturation at 95 °C for 10 min, followed by 40 cycles of 95 °C for 20 s, 60 °C for 30 s, 72 °C for 1 min, and a final extension at 72 °C for 10 min. The primers used for the specific genes are listed in Supplementary Table 3.

## Electron microscopic analysis

Freshly dissected colon samples were fixed in 2.5% glutaraldehyde at 4 °C for 24 hrs. After rinsing three times with PBS buffer for 15 min each, the samples were further fixed with 1% $OsO_4$ solution (Ted Pella Inc., 012103) for 2 h. Following fixation, the samples were washed with PBS and dehydrated through a graded ethanol series (30%, 50%, 70%, 80%, 90%, and 100%), with each step lasting 15 min. For transmission electron microscopic (TEM) analysis, samples were embedded in EMBed 812 (SPI, 90529-77-4), sectioned at 70-90 nm using a Leica UC7 ultramicrotome, and mounted on copper grids (EMCN. Co., BZ100205a). Sections were stained with 2% uranyl acetate for 15 min, followed by lead citrate staining for 10 min. Grids were examined using a HITACHI HT7800 transmission electron microscope.

For scanning electron microscopic (SEM) analysis, samples were dried using a critical point dryer (Leica Microsystems, EM CPD300). The dried samples were mounted on aluminum stubs with conductive carbon adhesive tape (Ted Pella, 16084-2), sputter-coated with a thin layer of gold using an ion sputter coater (HITACHI, E-1010), and analyzed using a HITACHI SU8010 scanning electron microscope.

## Measurement of cytokine levels

Cytokine levels in colon homogenates were measured by the enzyme-linked immunosorbent assay (ELISA), using the following kits: Mouse TNF-α ELISA (Beyotime, PT512), mouse IL-6 ELISA (Beyotime, PI326), and mouse IL-10 ELISA (Beyotime, PI523), according to manufacturers' instructions. Cytokine levels were normalized to the protein concentration of the tissue lysate and presented as picograms or nanograms per milligram (pg or ng/mg).

## Statistical analysis

All the replicate experiments were repeated at least three times. For morphological analyses, at least three biological replicates were performed per group. Statistical analyses were performed using the GraphPad Prism software (Version 8.0.2) and OriginPro (Version 9.0.0). Survival curves were estimated using the Kaplan–Meier method, and the resulting curves were compared using the log-rank test. An unpaired two-tailed Student's t-test was used to assess the significance between two groups of normally distributed data. For data that were not normally distributed, an unpaired two-tailed Mann–Whitney test was employed. Comparisons between multiple groups with a single fixed factor were performed using ordinary one-way ANOVA or Kruskal-Wallis tests, followed by Tukey's or Dunn's post hoc tests, as indicated in the figure legends. Statistical significance was defined as $p < 0.05$.*, $p < 0.05$; **, $p < 0.01$; ***, $p < 0.001$, ns indicates non-significant.

## Reporting summary

Further information on research design is available in the Nature Portfolio Reporting Summary linked to this article.

# Data availability

The data, analytical methods, and study materials will be made available to other researchers. Further information and requests for resources and reagents should be directed to Yajun Xie (yjxie@cq-mu.edu.cn). Source data are provided with this paper.

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

## Acknowledgements

We would like to thank Prof. Zhimin Lu and Daqian Xu from Zhejiang University and Prof. Zehan Hu from Shanghai Jiao Tong University for their suggestions. This work was funded by the following bodies: National Natural Science Foundation of China (Grant numbers 82030065, 32270834, and 31701218), National Key R&D Program from Ministry of Science and Technology of China (2023YFE0113500), Basic Science and Frontier Technology Research Program of Chongqing Science and Technology Commission (Grant number cstc2021jcyj-msxmX0312, CSTB2023NSCQ-MSX0318). M.P. is supported by funds from Karolinska Institutet, the Swedish Research Council (Vetenskaps-rådet, 2023-03095), the Ragnar Söderberg Foundation (M21/17), and Cancerfonden (21 1620Pj).

## Author contributions

Z.Z. Conceptualization, Resources, Data curation, Software, Formal analysis, Performing experiments, Visualization, Methodology, Writing original draft. Y.X. Conceptualization, Resources, Performing experiments, Supervision, Funding acquisition, Project administration, Writing & editing manuscript. Q.Y. Supervision, Project administration, Writing review & editing. J.L. Data curation, Software, Formal analysis, Methodology, Writing original draft. L.Y. Data curation. R.W. Data curation. J.C. Data curation. X.L. Data curation. X.F. Data curation. S.Y. Formal analysis. Z.P. Formal analysis. M.P. Scientific Feedback, Writing & editing manuscript. Q.Z. Conceive and design the paper, Supervision, Funding acquisition.

## Competing interests

The authors declare no competing interests.
