## [Transparent Peer Review file · Nature Communications]

PEAK1 maintains tight junctions in intestinal epithelial cells and resists colitis by inhibiting autophagy-mediated ZO-1 degradation

Corresponding Author: Professor Yajun Xie

Version 0:

Reviewer comments:

Reviewer #1

(Remarks to the Author)

This study demonstrates the highly novel finding for a role for the pseudokinase PEAK1 in the maintenance of intestinal barrier integrity. The authors also characterise in impressive detail the molecular mechanisms by which this occurs and demonstrates a role for autophagy in tight junction regulation.

The experiments in general support the conclusions drawn by the authors.

Some comments below that will need to be addressed to further strengthen the conclusions drawn:

- Given the disordered microvilli reported in ZO-1 deficient mice, was this also observed in the PEAK1-deficient mice? It does not appear to be the case in Fig. 1j.
- In addition to ZO-1 and occludin, claudins also constitute a key component of the tight junction, yet no reference is made to whether claudins are destabilised, mislocalised or degraded when PEAK1 is deleted and ZO-1 levels are reduced. It would be of relevance to include analysis of claudins as well.
- In Fig. 1e, the orientation of the section is hard to decipher based on the images provided. It would be helpful to include a zoomed-out image, with corresponding H&E-stained sections. Only one section is presented, and it would be good to see multiple sections (in Supplementary). Also, unless I have misinterpreted, why is there ZO-1 staining, particularly in the WT sections, at the basal end of the IECs given their apical localisation?
- It would be useful to demonstrate that the PEAK1 Y724F mutant fails to co-immunoprecipitate with ZO1 as part of Fig. 3g given the availability of the mutant Caco-2 cell line generated by CRISPR/Cas0 gene-editing.
- A more representative image of a larger section needs to be presented for Fig. 6d.

Reviewer #2

(Remarks to the Author)

This manuscript finds that deletion of PEAK1 (pseudopodium enriched atypical kinase 1) leads to reduced expression of tight junction proteins ZO-1 and occludin. This regulation appears to be post-transcriptional, as mRNA expression is unchanged. The investigators continue to determine that ZO-1 binds to PEAK1 and identify a specific phosphorylatable site on PEAK1 regulates binding to and stabilization of ZO-1. Finally, they confirm previous work showing that CSK binds to PEAK1.

The in vitro protein work is extensive and, with the exception of the studies involving CSK, carefully performed. However, the analyses are incomplete. More importantly, the in vivo biology with mice and studies using cell lines have substantial shortcomings. This undercuts the conclusion, stated in the title, that PEAK1 maintains tight junctions. Further work is also required to demonstrate that PEAK1 inhibits autophagy-mediated ZO-1 degradation and that this leads to resistance to colitis (as a minor point, enteritis, inflammation of the small intestine, is not studied here).

Specific concerns:

1. Much of the data, including western blots and morphology, lacks quantitative analysis. This is needed.
2. Data quality should be improved in several places.
 - a. Fig 1b the immunostains look like a brown blush rather than specific staining.
 - b. Fig 2b need statistical analysis. There seems to be a GFP peak in the controls.
 - c. Fig 7e shows H&E images of proximal colon in WT and distal colon in KO. DSS induces distal colitis with a skip lesion (see PMID17200722 Fig 1A).
3. A previous report concludes that activation of autophagy prevents ZO-1 degradation and reduces disease induced by DSS (PMID32952341)
4. Better characterization of the Peak1 KO mice is needed. What is ZO1 distribution at baseline. Is there junction tortuosity? How is epithelial cell turnover affected? Are there changes in other cell types, such as immune cells?
5. The previous in vivo study of ZO1 KO (ref 12, Wei-Ting et al) found that macromolecular permeability was only slightly altered by ZO-1 (Tjp1) KO in the intestine. Like Peak1 KO mice, intestinal epithelial Tjp1 KO mice display increased sensitivity to DSS. However, those authors concluded that the defect was in epithelial repair, not sensitivity to damage. That may be the case with Peak1 KO mice, but further work is needed to determine this.
6. Transmission electron microscopy cannot be used to image tight junction strands. Freeze fracture electron microscopy is required. In addition, the subtle difference (seen in the insets) between the images is within expected variation between cells. Quantitative morphometry would be required to show that a difference exists. Even if the difference claimed is real, no rigorous studies have shown that the space between adjacent cell membranes at tight junctions relates to function(although some studies used this measurement, none have validated it.
7. PEAK1 Y724F should be studied in PEAK KO cells. Are the results shown due to a dominant negative effect?
8. Affinities of protein interactions should be assessed quantitatively.
9. Increased FITCdextran flux could indicate changes in tight junction barrier but may also result from low grade injury. This concern is accentuated by the fact that the Peak1 KO is a global KO. Permeability should be assessed additional larger probes, as described PMID34048016.
10. Caco2 are known to have significant clonal variation. How many KO clones were examined? PEAK1 expression should be used to confirm the changes in Caco2 induced by KO.
11. PEAK1 is known to regulate the cytoskeleton, and the cytoskeleton is known to regulate tight junctions via ZO1. Some papers on this topic have linked Src, myosin, ZO1, and increased tight junction tortuosity. One showed that myosin inhibition reversed tortuosity (PMID16638813). This should be considered as potential mediator of PEAK1 effects.
12. PCR and western blot analyses normalize to universally expressed housekeeping genes. The concurrent reductions in occludin, ZO1, and PEAK1 could all simply reflect reduced numbers of epithelial cells relative to inflammatory cells. Normalization should be to epithelial proteins whose expression on a per cell basis does not change.
13. Significant conclusions rely on drugs that can have many other effects, including bafilomycin, PP2, and rapamycin. More detailed examination of their effects on src, PEAK1, and ZO1 and other potential explanations for the effects of these drugs should be considered.
14. Many articles linking tight junctions to src, ZO1 phosphorylation, and autophagy are not considered and undercut some of the novelty of this work. Examples include PMID17446308, PMID11352822, PMID23064762, PMID36999034, PMID34964704, PMID25616664, PMID31542404.

Reviewer #3

(Remarks to the Author)

General Comments

In the manuscript NCOMMS-24-56544-T "PEAK1 maintains tight junctions in intestinal epithelial cells and resists enteritis by inhibiting autophagy-mediated ZO-1 degradation", the authors show a new role of PEAK1 interaction with ZO-1 in the amino acid range of 713 – 731 with phosphorylation of Y724 to be necessary for the interaction.

I found this manuscript to be very well written and only have a few minor revisions. My comments are specific to the proteomic portion of the manuscript.

Specific Comments

1. This manuscript describes two proteomic experiments. The first is an immunoprecipitation and the second is to identify the site of phosphorylation. The method is very generic and does not adequately describe each experiment. The method second should describe separately how the IP was done and then how the phosphorylation determination was done. For example, on page 7 where the IP results are shown, it is not clear if the mass spectrometry data was from a gel band or from the whole IP eluting proteins. Second, it is not clear in the manuscript what the phosphorylation mass spectra is from. I think it is a gel band that was lined up using the western blot data, but it is not clear.
2. The method needs to be explicit to what the post-translational modification settings were. It just states "and other interested post-translational modifications were specified".
3. On page 27, line 567 – the author says, "the isolated peptides were subjected to Nano source". That is nonsensical. It should be "subjected to nanospray on a (what was the nanospray source used)"
4. The actual mass spectrometry details need to be added. Here is an example of what I publish in my lab
The scan sequence of the mass spectrometer was based on the original TopTen™ method; the analysis was programmed for a full scan recorded between 375 – 1575 Da at 60,000 resolution, and a MS/MS scan at resolution 15,000 to generate product ion spectra to determine amino acid sequence in consecutive instrument scans of the fifteen most abundant peaks in the spectrum. The AGC Target ion number was set at 3e6 ions for full scan and 2e5 ions for MS2 mode. Maximum ion

injection time was set at 50 ms for full scan and 55 ms for MS2 mode. Micro scan number was set at 1 for both full scan and MS2 scan. The HCD fragmentation energy (N)CE/stepped NCE was set to 28 and an isolation window of 4 m/z. Singly charged ions were excluded from MS2. Dynamic exclusion was enabled with a repeat count of 1 within 15 seconds and to exclude isotopes. A Siloxane background peak at 445.12003 was used as the internal lock mass.

Version 1:

Reviewer comments:

Reviewer #1

(Remarks to the Author)

I am satisfied with the revisions made by the authors which have significantly improved the robustness of the conclusions and the experimental data. I'd like to commend the authors on the efforts made to address all comments.

Reviewer #2

(Remarks to the Author)

The results are largely correlative and technical issues remain. As an example of the latter, the new higher power EMs show the adherens junctions, not the tight junctions. Finally, it would be critical to show that ZO-1 overexpression prevents the effect of PEAK1 KO. Correlation with reduced ZO-1 levels is not sufficient to infer that ZO-1 downregulation is responsible for the changes observed.

Reviewer #3

(Remarks to the Author)

Thank you for addressing all of this reviewers points. The manuscript is much stronger.

Version 2:

Reviewer comments:

Reviewer #1

(Remarks to the Author)

I have had a close look at the author's comments and the experiments they have undertaken to address Reviewer 2's comments.

I strongly believe that the authors have fully addressed the concerns raised by Reviewer 2 (at least from my perspective) and that the results robustly address these.

I personally have no issue with agreeing that the authors have satisfactorily addressed what has been asked of them by Reviewer 2.

Reviewer #2

(Remarks to the Author)

Reviewer #4

(Remarks to the Author)

The authors properly addressed the remaining questions of the reviewer. I have no additional questions or comments.

We sincerely thank the reviewer's insightful comments. Addressing them has helped us improve the manuscript for your review. Each of the concerns raised by the three reviewers has been addressed and we provide a point-by-point response below, highlighted in blue. Modifications in the revised manuscript are highlighted in yellow.

REVIEWER COMMENTS

Reviewer #1 (Remarks to the Author):

This study demonstrates the highly novel finding for a role for the pseudokinase PEAK1 in the maintenance of intestinal barrier integrity. The authors also characterise in impressive detail the molecular mechanisms by which this occurs and demonstrates a role for autophagy in tight junction regulation.

The experiments in general support the conclusions drawn by the authors.

Some comments below that will need to be addressed to further strengthen the conclusions drawn:

-Given the disordered microvilli reported in ZO-1 deficient mice, was this also observed in the PEAK1-deficient mice? It does not appear to be the case in Figure 1j.

Response: Yes, this is correct. Based on our initial results (Figure 1j), we did not observe obvious microvilli abnormalities in *Peak1*^{-/-} mice. To further confirm this, we have added new data in the revised manuscript, showing that *Peak1*^{-/-} mice, unlike ZO-1-deficient mice, do not exhibit disordered microvilli.

Updated Figure 1j: Transmission electron microscopy reanalysis of the tight junctions of colonic epithelial cells, using a 100000 times magnification, more clearly shows that PEAK1 deficiency does not significantly affect microvilli growth.

New Supplementary Figure 1r & Unpublished Figure 1: Scanning electron microscopies of the colon and small intestine reveal no significant morphological differences in *Peak1*^{-/-} mice.

Figure 1j

Supplementary Figure 1r

Unpublished Figure 1

- In addition to ZO-1 and occludin, claudins also constitute a key component of the tight junction, yet no reference is made to whether claudins are destabilised, mislocalised or degraded when PEAK1 is deleted and ZO-1 levels are reduced. It would be of relevance to include analysis of claudins as well.

Response: Thank you for suggesting these important additions. We conducted additional western blot experiments to assess Claudin 2 and Claudin 4 expression in the colons of DSS-treated WT and *Peak1*^{-/-} mice (Figure 1a, c, d). We also performed immunofluorescence to analyze their localization (Supplementary Figure 1q). Our results show that, in addition to ZO-1 and Occludin, Claudin 2 levels-but not Claudin 4-are reduced in both DSS-treated WT as well as untreated *Peak1*^{-/-} mice. We believe that Claudin 2, like Occludin instability, could be linked to loss of ZO-1, as previously reported (PMID:29187366; PMID:10601346); this is now mentioned in the text.

Figure 1a

Figure 1c

Figure 1d

Supplementary Figure 1q

- In Figure 1e, the orientation of the section is hard to decipher based on the images provided. It would be helpful to include a zoomed-out image, with corresponding H&E-stained sections. Only one section is presented, and it would be good to see multiple sections (in Supplementary). Also, unless I have misinterpreted, why is there ZO-1 staining, particularly in the WT sections, at the basal end of the IECs given their apical localisation?

Response: We have added multiple zoomed-out corresponding H&E-stained colon sections from WT and *Peak1*^{-/-} mice in the revised manuscript to clarify image orientation and apical/basal locations (Figure 1e, Supplementary Figure 1n-p). Additionally, we used a different primary antibody conjugated with fluorescent tags to better visualize ZO-1 in IECs. While ZO-1 is present in both the basal and apical regions in WT sections (Figure 1e, Supplementary Figure 1o), its expression is notably higher at the apical surface, as expected. This was included in revision.

Figure 1e

Supplementary Figure 1n

Supplementary Figure 1o

Supplementary Figure 1p

-It would be useful to demonstrate that the PEAK1 Y724F mutant fails to co-immunoprecipitate with ZO1 as part of Figure 3g given the availability of the mutant Caco-2 cell line generated by CRISPR/Cas0 gene-editing.

Response: We appreciate the reviewer's suggestion to include this important experiment, which we have now incorporated into the revised manuscript (Figure 3h). By immunoprecipitating ZO-1 and immunoblotting for PEAK1 in cell lysates from WT and Y724F mutant Caco-2 cells, we demonstrate that the PEAK1 Y724F mutant fails to co-immunoprecipitate with ZO-1.

Figure 3h

- A more representative image of a larger section needs to be presented for Figure 6d.

Response: We have replaced the original image with a larger, more representative section showing ZO-1 colocalization with LC3B-positive autophagosomes (new Figure 6d). While the image still covers only two cells, it includes more areas expressing both ZO-1 and autophagosomes. This representation is consistent with our findings across multiple sections/cells, as shown in other autophagy immunofluorescence analyses (Figure 6j, 6l, 6n, n numbers 80 cells analyzed in each experiment). We believe an even larger image may reduce clarity in visualizing staining and colocalization, but we are happy to provide one if the reviewer finds it necessary.

Figure 6d

Reviewer #2 (Remarks to the Author):

This manuscript finds that deletion of PEAK1 (pseudopodium enriched atypical kinase 1) leads to reduced expression of tight junction proteins ZO-1 and occludin. This regulation appears to be post-transcriptional, as mRNA expression is unchanged. The investigators continue to determine that ZO-1 binds to PEAK1 and identify a specific phosphorylatable site on PEAK1 regulates binding to and stabilization of ZO-1. Finally, they confirm previous work showing that CSK binds to PEAK1.

The in vitro protein work is extensive and, with the exception of the studies involving CSK, carefully performed. However, the analyses are incomplete. More importantly, the in vivo biology with mice and studies using cell lines have substantial shortcomings. This undercuts the conclusion, stated in the title, that PEAK1 maintains tight junctions. Further work is also required to demonstrate that PEAK1 inhibits autophagy-mediated ZO-1 degradation and that this leads to resistance to colitis (as a minor point, enteritis, inflammation of the small intestine, is not studied here).

We thank the reviewer for their careful insight and recommendations. Each concern is addressed in a point-by-point response below. Additionally, we have replaced the term "enteritis" with "colitis" throughout the manuscript for greater accuracy.

Specific concerns:

1. Much of the data, including western blots and morphology, lacks quantitative analysis. This is needed.

Response: We apologize for this important omission in the original manuscript. We have now included the quantitative analysis of western blots and morphological studies in the revised version.

2. Data quality should be improved in several places.

a. Fig 1b the immunostains look like a brown blush rather than specific staining.

Response: Our PEAK1 antibody and staining are specific, as demonstrated by the complete lack of signal in sections from PEAK1-KO mice (added to Figure 1b). We have also included brighter IHC images to better visualize the localization and expression of PEAK1 in the colon during DSS treatment (Figure 1b).

Figure 1b

b. Fig 2b need statistical analysis. There seems to be a GFP peak in the controls.

Response: We have repeated the colocalization studies of GFP-PEAK1 and ZO-1, addressing the concern regarding the GFP peak in the controls. The revised manuscript now includes more representative images and statistical analysis (Figure 2b and 2c).

Figure 2b

Figure 2c

c. Fig 7e shows H&E images of proximal colon in WT and distal colon in KO. DSS induces distal colitis with a skip lesion (see PMID17200722 Fig 1A).

Response: We apologize for the inconsistency. We have now included H&E images and corresponding histological scores for both the proximal and distal colon in WT and PEAK1^{-/-} DSS-treated mice (Figure 7e and 7f). As the reviewer noted and as is well-established in the field, our DSS model primarily induces distal colitis, with marked differences between WT and *Peak1^{-/-}* mice. The proximal colon is only mildly affected by DSS, regardless of PEAK1

expression.

Figure 7e

Figure 7f

3. A previous report concludes that activation of autophagy prevents ZO-1 degradation and reduces disease induced by DSS (PMID32952341)

Response : In the cited study by Hang-Hai Pan et al, they demonstrate that inducing autophagy with resveratrol ameliorated the DSS- induced colitis in mice. By histology, they show a more intact intestinal mucosal barrier in line with a milder colitis phenotype, that expressed, in consequence of retaining a more intact intestinal barrier, higher levels of ZO-1, as determined by immunohistochemistry. However, in addition to IHC not being a quantitative method, there are other limitations in this study that prevent a direct conclusion that autophagy protects against ZO-1 degradation to maintain tight junctions and reduce colitis. Resveratrol is a clinically validated antioxidant and anti-inflammatory agent and has been demonstrated as a non-canonical autophagy inducer in breast cancer cells (PMID: 28945937), and the effects of resveratrol on LC3II levels in their reports were mild, raising questions about the specificity of their findings. Additionally, autophagy influences several intestinal processes, including the secretory lineage (PMID: 23216414; PMID: 33906557), which could contribute to colitis protection. In our opinion the correlations between ZO-1 IHC levels, histology, and colitis scores in that study are interesting but not sufficient to establish causality. We instead provide extensive conclusive evidence using both the classical autophagy activator rapamycin as well as the potent inhibitor bafilomycin that autophagy directly regulates the stability of ZO-1 (Figure 6a and 6b, supplementary Figure 7 b-d).

Figure 6a

Figure 6b

Supplementary Figure 7b

Supplementary Figure 7c

Supplementary Figure 7d

4. Better characterization of the Peak1 KO mice is needed. What is ZO1 distribution at baseline. Is there junction tortuosity? How is epithelial cell turnover affected? Are there changes in other cell types, such as immune cells?

Response: We conducted a more thorough analysis of baseline (unchallenged) *Peak1*^{-/-} mice, which now includes body weight (Supplementary Figure 8a), colon length (Supplementary Figure 8b), H&E histology of distal colons and intestine (Supplementary Figure 8c-d), proliferation of colonic epithelial cells (Supplementary Figure 8e), immune cell numbers (F4/80+ macrophages, Figure 7h), and key inflammatory-related cytokines (Figure 7i-k). Additionally, we performed scanning electron microscopy to assess microvilli growth of colon and intestine (Supplementary Figure 1r, Unpublished figure 1).

In all these parameters, *Peak1*^{-/-} mice were indistinguishable from WT littermates, except for slightly increased level of the anti-inflammatory cytokine IL-10. Whether this change is linked to the increased leak permeability observed at baseline will require further investigation. Overall, our deeper characterization of *Peak1*^{-/-} mice shows that depletion of PEAK1 is insufficient to spontaneously cause colitis.

To determine the distribution of ZO-1 we performed new immunofluorescence studies with primary ZO-1 antibody conjugated to fluorescent tags (Figure 1e and Supplementary Figure 1n). Tight junctions are mainly distributed at apical region of IECs, the baseline of IECs are mainly adhesive connections, bridge particles and other structures. ZO-1 is expressed in multiple locations such as cytoplasm and cortex of cells.

Supplementary Figure 8a

Supplementary Figure 8b

Supplementary Figure 8c

Supplementary Figure 8d

Supplementary Figure 8e

Figure 7h

Figure 7i

Figure 7j

Figure 7k

Supplementary Figure 1r

Unpublished Figure 1

Figure 1e

Supplementary Figure 1n

5. The previous in vivo study of ZO1 KO (ref 12, Wei-Ting et al) found that macromolecular permeability was only slightly altered by ZO-1 (Tjp1) KO in the intestine. Like Peak1 KO mice,

intestinal epithelial Tjp1 KO mice display increased sensitivity to DSS. However, those authors concluded that the defect was in epithelial repair, not sensitivity to damage. That may be the case with Peak1 KO mice, but further work is needed to determine this.

Response: We apologize for not more clearly addressing the role of PEAK1 in damage versus intestinal repair. During the first 7 days of DSS treatment (the damage phase), there was no significant difference in body weight loss between Peak1^{-/-} and WT mice. However, the difference became apparent once water was reintroduced, marking the recovery phase when DSS-treated mice begin to rebuild their intestinal epithelial barrier. The severe defect in body weight recovery in Peak1^{-/-} mice compared to WT mice suggests that PEAK1, like ZO-1, has crucial functions during regeneration rather than damage (Figure 7a).

To further support this conclusion, we performed TUNEL and Ki67 immunohistochemistry analyses of colons from Peak1^{-/-} mice with or without DSS treatment (Supplementary Figure 8e and 8f). As expected, DSS-treated WT mice showed decreased numbers of apoptotic cells and increased proliferative Ki67-positive cells in crypts, indicating intestinal repair. In contrast, PEAK1 deletion in colonic epithelial cells diminished proliferation and enhanced apoptosis after DSS treatment (Supplementary Figure 8e and 8f), supporting that PEAK1 deficiency, like ZO-1 deficiency, impairs IEC recovery following DSS-induced damage. These results are now included and better described in the manuscript.

Figure 7a

Supplementary Figure 8e

Supplementary Figure 8f

6. Transmission electron microscopy cannot be used to image tight junction strands. Freeze fracture electron microscopy is required. In addition, the subtle difference (seen in the insets) between the images is within expected variation between cells. Quantitative morphometry would be required to show that a difference exists. Even if the difference claimed is real, no rigorous studies have shown

that the space between adjacent cell membranes at tight junctions relates to function (although some studies used this measurement, none have validated it).

Response: We made numerous efforts to improve our analysis as recommended by the reviewer and reached out to several prestigious institutions, including Peking University, Tsinghua University, and the Chinese Academy of Sciences, for assistance with the requested freeze-fracture electron microscopy study. Unfortunately, due to recent technological and instrument updates, none of these institutions retained the complete sample preparation system required for this technique.

As an alternative, to better observe the effect of PEAK1 deficiency on tight junctions, we conducted transmission electron microscopy again, magnifying the images 100,000 times. The resulting structure clearly showed that the cell membranes of adjacent IECs in WT mice exhibited an interlaced pattern (some fusion points on the membrane between two IECs cells), whereas in *Peak1*^{-/-} mice, the cell membranes were parallel to each other and lacked the interlaced structure (Figure 1j). DSS treatment further increased the intercellular gaps between adjacent IECs in *Peak1*^{-/-} mice (Figure 7g). We hope the reviewer finds our new efforts and results more satisfactory.

Figure 1j

Supplementary Figure 2I

Figure 7g

Supplementary Figure 8g

7. PEAK1 Y724F should be studied in PEAK KO cells. Are the results shown due to a dominant negative effect?

Response: We have significantly expanded our studies using the PEAK1 Y724F mutant. First, we immunoprecipitated ZO-1 and immunoblotted for PEAK1 in cell lysates from WT and Y724F mutant Caco-2 cells, confirming that the PEAK1 Y724F mutant fails to co-immunoprecipitate with ZO-1 (Figure 3h).

Figure 3h

We then performed several functional studies using PEAK1 Y724F overexpression in PEAK1 KO cells. In Caco-2 cells depleted of PEAK1, overexpression of PEAK1 Y724F failed to restore the structural and functional impairments of tight junctions caused by PEAK1 deficiency, including increased tight junction tortuosity, decreased transepithelial electrical resistance, elevated FITC-dextran permeability, and reduced ZO-1 and Occludin levels (Supplementary Figure 4i-4n).

Supplementary Figure 4i

Supplementary Figure 4j

Supplementary Figure 4k

Supplementary Figure 4l

Supplementary Figure 4m

Supplementary Figure 4n

8. Affinities of protein interactions should be assessed quantitatively.

Response: To better address this point, we have expressed and purified GFP-ZO-1, PEAK1, and PEAK1 Y724 proteins in eukaryotic 293T cells and used pulldown assays to investigate the interactions between PEAK1 or FC-PEAK1 Y724 proteins and GFP-ZO-1 at serial concentrations (see new Figure 2i).

Figure 2i

9. Increased FITC-dextran flux could indicate changes in tight junction barrier but may also result from low grade injury. This concern is accentuated by the fact that the Peak1 KO is a global KO. Permeability should be assessed additional larger probes, as described PMID34048016.

Response: Thank you for suggesting this important experiment. We have now conducted additional in vivo permeability studies using different-sized probes: a larger Rhodamine-Dextran (70kDa), which indicates unrestricted permeability pathways (tight junction-independent), and a smaller creatinine (113Da), which indicates pore permeability pathways, that like 4kDa FITC-dextran, is tight junction-dependent.

Compared to WT littermates, *Peak1*^{-/-} mice showed increased intestinal permeability to 4kDa FITC-dextran and 113Da creatinine (Figure 1k, m), but not 70kDa Rhodamine-dextran (Figure 1l). This confirms that the increased permeability in the absence of PEAK1 is tight-junction dependent and suggests no low-grade injury or upregulation of unrestricted permeability pathways.

Figure 1k

Figure 1l

Figure 1m

10. Caco2 are known to have significant clonal variation. How many KO clones were examined? PEAK1 expression should be used to confirm the changes in Caco2 induced by KO.

Response: To evaluate potential effects driven by clonal variation, we constructed PEAK1 knockout Caco-2 cells using CRISPR-Cas9 and selected clones for subsequent analysis. In all four clones, we confirmed the absence of PEAK1, which consistently and markedly reduced ZO-1 and Occludin levels (Supplementary Figure 2a). Additionally, these findings were replicated using specific siRNAs to knock down PEAK1 in Caco-2 cells (Supplementary Figure 2b) and in colon lysates from multiple *Peak1*^{-/-} mice (Fig. 1c). Therefore, we are confident that Caco-2 clonal variation did not significantly impact our results.

Supplementary Figure 2a

Supplementary Figure 2b

Figure 1c

11. PEAK1 is known to regulate the cytoskeleton, and the cytoskeleton is known to regulate tight junctions via ZO1. Some papers on this topic have linked Src, myosin, ZO1, and increased tight junction tortuosity. One showed that myosin inhibition reversed tortuosity (PMID16638813). This should be considered as potential mediator of PEAK1 effects.

Response: We agree with the reviewer's comment. Given the known function of PEAK1 in actin cytoskeleton organization, it is important to consider whether this mechanism could also be regulating tight junction permeability and ZO-1. In the early stages of this project, we tested the levels of Myosin Light Chain (MLC) phosphorylation and MLC Kinase (MLCK) in PEAK1 knockout Caco-2 cells. Since PEAK1 absence did not markedly alter MLCK expression or MLC phosphorylation (Unpublished Figure 2 below), and neither MLCK nor

MLC were found to interact with PEAK1 in our immunoprecipitation assays (Supplementary Figure 3b), we did not continue exploring a potential role for myosin/cytoskeleton in mediating PEAK1 functions in tight junctions.

That said, we cannot completely exclude the possibility that PEAK1 might influence the cytoskeleton or other molecular mechanisms not yet explored, which could also affect ZO-1 and tight junctions. This could be an interesting avenue for future research. However, based on our extensive and consistent experimental data across different models and methodologies, we believe it is unlikely that the observed phenotypes are entirely independent of the newly discovered PEAK1-ZO-1 direct interaction in the regulation of ZO-1 autophagy. We have now included a shorter version of this relevant discussion in the revised manuscript.

12. PCR and western blot analyses normalize to universally expressed housekeeping genes. The concurrent reductions in occludin, ZO1, and PEAK1 could all simply reflect reduced numbers of epithelial cells relative to inflammatory cells. Normalization should be to epithelial proteins whose expression on a per cell basis does not change.

Response: The point raised by the reviewer is valid, particularly concerning the western blot analysis of DSS samples, where the ratio of epithelial to immune cells in the intestines changes significantly. Notably, in unchallenged *Peak1*^{-/-} mice, we observe dramatic changes in ZO-1 and Occludin levels, but we do not see alterations in the structure, length, or histology of the colon, nor any obvious signs of inflammation or immune cell infiltration (as explained in point 5). This suggests that if the epithelial/immune cell ratio is indeed altered in *Peak1*^{-/-} mice, it would only be a very mild change and unlikely to be responsible for the strong reductions in Occludin, ZO-1, and PEAK1 levels.

To further validate our western blot data, we also blotted for additional epithelial proteins, including Claudin 2 and Claudin 4. Total Claudin 4 levels in colon and IECs from *Peak1*^{-/-} mice did not differ significantly from WT mice when normalized to Ponceau staining or housekeeping genes (Figure 1c and d). We repeated the quantification of ZO-1, PEAK1, and Occludin levels in the intestine of *Peak1*^{-/-} mice, normalizing to Claudin 4, and obtained the

same results: Peak1 deletion led to markedly lower levels of ZO-1, Occludin, and Claudin 2. These specific effects of PEAK1 deletion on ZO-1 and Occludin and Claudin 2 were also observed in four independent Caco-2 PEAK1 knockout clones and Caco-2 knockdown cells (where no other cells are present and where E-cadherin, another epithelial protein, was unchanged) (Supplementary Figure 2a) . To validate our qPCR data, we provide now absolute RNA values obtained by RNA sequencing analysis of the colon from WT and *Peak1*^{-/-} mice (Supplementary Figure 1f-k) confirm that PEAK1 deficiency does not alter the mRNA levels of tight junction proteins such as ZO-1, Claudin 2, Claudin 4, and Occludin.

Figure 1c

Figure 1d

Supplementary Figure 2a

Supplementary Figure 1f-1k

13. Significant conclusions rely on drugs that can have many other effects, including bafilomycin, PP2, and rapamycin. More detailed examination of their effects on src, PEAK1, and ZO1 and other potential explanations for the effects of these drugs should be considered.

Response: We agree with the reviewer that drug side effects and toxicity must always be considered and carefully evaluated. To minimize the risk of such effects, we used only well-characterized drugs, protocols, and doses, including dose-response studies across different cell types (Supplementary Figure 7b-d, g-h). As per the reviewer's request, we performed additional western blot analyses to specifically assess the effects of PP2, bafilomycin, and rapamycin treatments on Src, PEAK1, and ZO-1:

- i) PP2 treatment decreased p-Src (Y416) and ZO-1 levels but did not affect PEAK1 or Src/ERK (Figure 3c).
- ii) Bafilomycin A1 treatment increased ZO-1 and Occludin levels, with no impact on PEAK1,

Src, or CSK (Unpublished Figure 3).

iii) Rapamycin treatment decreased ZO-1 and Occludin levels, without influencing PEAK1, Src, or CSK (Unpublished Figure 4).

14. Many articles linking tight junctions to src, ZO1 phosphorylation, and autophagy are not considered and undercut some of the novelty of this work. Examples include PMID17446308, PMID11352822, PMID23064762, PMID36999034, PMID34964704, PMID25616664, PMID31542404.

Response: We agree with the reviewer that previous reports linking Src, ZO-1 phosphorylation, and autophagy to tight junction regulation have weakened the novelty of our work to some extent. However, the specific mechanism of ZO-1 stability remains unresolved in the literature. Our study presents PEAK1 as a novel tight junction protein that positively

regulates Src activity and prevents ZO-1 degradation, thereby reshaping tight junction dynamics. In response to the reviewer's comment, we have now incorporated relevant literature, including some of the cited studies, to better contextualize the novelty and impact of our findings in the revised manuscript.

Reviewer #3 (Remarks to the Author):

General Comments

In the manuscript NCOMMS-24-56544-T “PEAK1 maintains tight junctions in intestinal epithelial cells and resists enteritis by inhibiting autophagy-mediated ZO-1 degradation”, the authors show a new role of PEAK1 interaction with ZO-1 in the amino acid range of 713 – 731 with phosphorylation of Y724 to be necessary for the interaction.

I found this manuscript to be very well written and only have a few minor revisions. My comments are specific to the proteomic portion of the manuscript.

Specific Comments

1. This manuscript describes two proteomic experiments. The first is an immunoprecipitation and the second is to identify the site of phosphorylation. The method is very generic and does not adequately describe each experiment. The method second should describe separately how the IP was done and then how the phosphorylation determination was done. For example, on page 7 where the IP results are shown, it is not clear if the mass spectrometry data was from a gel band or from the whole IP eluting proteins. Second, it is not clear in the manuscript what the phosphorylation mass spectra is from. I think it is a gel band that was lined up using the western blot data, but it is not clear.

Response: We apologize for the confusion. We revised the Methods section to clarify the methodology used in our proteomic experiments. The updated section provides a more detailed explanation of the experimental approach to ensure clearer understanding of the techniques employed.

2. The method needs to be explicit to what the post-translational modification settings were. It just states “and other interested post-translational modifications were specified”.

Response: In the revised version, we have now explicitly outlined the other specific post-translational modifications considered (phosphorylation) in our proteomic experiments, as requested.

3. On page 27, line 567 – the author says, “the isolated peptides were subjected to Nano source”. That is nonsensical. It should be “subjected to nanospray on a (what was the nanospray source used)”

Response: We have revised this description in the manuscript and now specify the type of nanospray source used in the experiment (EASY Spray).

4. The actual mass spectrometry details need to be added. Here is an example of what I publish in my lab

The scan sequence of the mass spectrometer was based on the original TopTen™ method; the analysis was programmed for a full scan recorded between 375 – 1575 Da at 60,000 resolution, and a MS/MS scan at resolution 15,000 to generate product ion spectra to determine amino acid

sequence in consecutive instrument scans of the fifteen most abundant peaks in the spectrum. The AGC Target ion number was set at 3e6 ions for full scan and 2e5 ions for MS2 mode. Maximum ion injection time was set at 50 ms for full scan and 55 ms for MS2 mode. Micro scan number was set at 1 for both full scan and MS2 scan. The HCD fragmentation energy (N)CE/stepped NCE was set to 28 and an isolation window of 4 m/z. Singly charged ions were excluded from MS2. Dynamic exclusion was enabled with a repeat count of 1 within 15 seconds and to exclude isotopes. A Siloxane background peak at 445.12003 was used as the internal lock mass.

Response: We appreciate the reviewer's suggestions and the shared template for guidance. We have now revised the manuscript to provide a more detailed description of the protein identification method used.

We sincerely thank all the reviewers for their insightful comments throughout the review process, which have helped us strengthen the manuscript. We are pleased that two reviewers have expressed full satisfaction with our revisions and have no further comments. The few remaining critiques raised by Reviewer #2 have been addressed with further experiments and in a point-by-point response below, with our response highlighted in blue. The corresponding modifications and additions in the manuscript are highlighted in yellow.

REVIEWER COMMENTS

Reviewer #1 (Remarks to the Author):

I am satisfied with the revisions made by the authors which have significantly improved the robustness of the conclusions and the experimental data. I'd like to commend the authors on the efforts made to address all comments.

Response: We are grateful to Reviewer #1 for the positive feedback. We are pleased to hear that our previous revisions have successfully addressed your comments. Thank you for your thorough review and constructive suggestions, which have helped improve the quality of our manuscript.

Reviewer #2 (Remarks to the Author):

The results are largely correlative and technical issues remain. As an example of the latter, the new higher power EMs show the adherens junctions, not the tight junctions. Finally, it would be critical to show that ZO-1 overexpression prevents the effect of PEAK1 KO. Correlation with reduced ZO-1 levels is not sufficient to infer that ZO-1 downregulation is responsible for the changes observed.

Response: Thank you again for your thorough review and valuable suggestions to improve the manuscript. We have performed the additional experiments you requested and have addressed your concerns about correlations and the proper identification of tight junctions.

The location and characteristics of tight junctions (TJs) are well established in the field (e.g., PMID: 13944428; 32891490; 19029935, 38712627, 22583600, 37186118, 36715075). In ultrathin sections, TJs are observed as epithelial intercellular junctions located at the most apical region of cell-cell contacts (see example pictures from publication -a). These junctions are characterized by the local fusion of adjacent cell membranes, a feature known as “kiss points” (see example pictures from publication -b). Adherens junctions (AJs), in contrast, are located just below the tight junctions and are distinguished by a larger intercellular space.

The regions we studied and analyzed throughout our research correspond to the structures of colon villi, tight junctions, and adherens junctions as described in the referenced papers and others (Figure 1), particularly the TJs, that show the characteristic kiss points (Figure 1, 100,000x magnified image).

Pictures in publication

Figure 1j

Loss-of-function approaches, such as those used in our studies, including knockdowns by siRNAs and shRNAs, and particularly gene knockouts, are essential for identifying the function of a gene. The phenotype observed after a knockout, rather than being merely a correlation, provides clear evidence of the gene's role. However, it remains important to determine whether ZO-1 is mediating all of the PEAK1 effects on tight junctions (TJs) or only some of them.

As requested, we performed two additional functional experiments by overexpressing GFP-ZO-1 in PEAK1-depleted Caco-2 cells and assessed the extent of rescue in tight junction morphology and function. Overexpression of GFP-ZO-1 completely recovered the increased tortuosity caused by PEAK1 deficiency in Caco-2 cells. Additionally, although the transendothelial electrical resistance (TEER) and FITC-Dextran permeability were rescued, they did not return to control levels (Supplementary Figure 2m-q). These new results have been integrated into Supplementary Figure 2 (Figures 2m-q).

Supplementary Figure 2

We then repeated the experiment, this time overexpressing GFP-ZO-1^{Δ1212-1217} in PEAK1-depleted Caco-2 cells to highlight the functional relevance of this newly identified LC3B binding domain. Overexpression of GFP-ZO-1^{Δ1212-1217} yielded the same results as GFP-ZO-1 full-length overexpression: it rescued the increased tortuosity caused by PEAK1 deficiency, as well as the TEER and FITC-Dextran permeability. These new results are added to Supplementary Figure 7 (panels i-m). Together, our new rescue experiments further emphasize, at a functional level, the essential and direct involvement of ZO-1 and its LC3 binding domain, in tight junction regulation by PEAK1.

Supplementary Figure 7

Reviewer #3 (Remarks to the Author):

Thank you for addressing all of this reviewers points. The manuscript is much stronger. We are grateful to Reviewer #3 for the positive feedback. Thank you for your thorough review and constructive suggestions, which have significantly contributed to enhancing the quality of our manuscript.